# The plastid cysteine synthase complex regulates ABA biosynthesis and stomatal closure in *Arabidopsis*

Sheng-Kai Sun [1], Nisar Ahmad[1,2], Hannah Callenius [1], Hala Rajab [1,3], Veli Vural Uslu[1,4], José Rey Cruz Cruz[1], Fang-Jie Zhao [5], Markus Wirtz [1] & Rüdiger Hell [1]

Global warming intensifies drought and high light stress periods, causing severe water loss and decreased crop yield. The phytohormone abscisic acid (ABA) is the dominant signal governing stomatal closure and water loss. Here, we uncover three signaling axes triggered by soil dehydration and high light stress converging on the dynamic assembly of the cysteine-synthase-complex in chloroplasts (pCSC). We show that pCSC assembly triggers ABA biosynthesis and stomatal closure in response to the soil-drying signals, sulfate (axis 1) and CLE25 (axis 2), and the high light-induced oxylipin OPDA (axis 3). Loss of the pCSC increases sensitivity to soil-drying and impairs high light-induced stomatal closure. Our findings uncover that the dynamic assembly of the pCSC acts as a sensor hub, integrating local and long-distance stress signals to promote stomatal closure by supplying cysteine for ABA biosynthesis in guard cells. We applied this knowledge to generate a soil-drying resilient plant showing no growth penalty.

As sessile organisms, flowering plants must adapt to rapidly changing environmental cues such as temperature and water supply. Stomata are tiny pores generated by guard cells that enable plants to take up carbon dioxide and control their body temperature through transpiration. Stomatal closure is one of the most rapid physiological responses to diverse abiotic stresses, including high light, soil-water limitation, and heat, enabling the plants to limit water loss[1,2]. Consequently, the tight control of the stomatal aperture is critical to optimizing plant survival and growth during drought stress reinforced by anthropogenically caused climate change. The phytohormone abscisic acid (ABA) is the key regulator of diverse stress responses, including drought-induced stomatal closure[3]. Upon soil water limitation and high light stress, ABA accumulates and induces transcription of response genes and stomatal closure via the ABA-signaling pathway[4,5]. Recently, the macronutrient sulfate and the peptide hormone CLA-VATA3/ESR-RELATED 25 (CLE25) have been shown to act as early root-to-shoot signals during drought stress that can induce stomatal closure by affecting ABA biosynthesis[6–9]. However, the primary signaling processes that trigger drought-induced ABA accumulation in guard cells remain unclear.

The last step of ABA biosynthesis is catalyzed by abscisic aldehyde oxidase 3 (AAO3)[10]. AAO3 is activated upon binding its sulfurylated molybdenum cofactor (MoCo)[11]. The sulfurylation of this essential cofactor is catalyzed by the sulfurtransferase ABA DEFICIENT 3 (ABA3), which accepts solely cysteine (Cys) as the sulfur donor[11]. Consequently, de novo Cys synthesis, which is the last step of the sulfate assimilation pathway, is required for ABA biosynthesis[6,9,12,13]. Furthermore, the application of sulfate or cysteine has been shown to trigger the transcriptional induction of *NCED3*, catalyzing the rate-limiting step in the synthesis of abscisic aldehyde, which AAO3 converts to ABA. Cys synthesis is initiated by the reduction of sulfate to sulfide, which occurs exclusively in plastids. The enzyme *O*-ACETYLSERINE(THIOL)

[1]Centre for Organismal Studies (COS), Heidelberg University, Heidelberg, Germany. [2]Department of Biotechnology and Genetic Engineering, Hazara University, Mansehra, Pakistan. [3]Peshawar Medical College, Riphah International University, Peshawar, Pakistan. [4]RLP AgroScience GmbH, Neustadt an der Weinstraße, Neustadt, Germany. [5]State Key Laboratory of Crop Genetics and Germplasm Enhancement and Utilization, College of Resources and Environmental Sciences, Nanjing Agricultural University, Nanjing, China. ✉e-mail: markus.wirtz@cos.uni-heidelberg.de; ruediger.hell@cos.uni-heidelberg.de

LYASE (OAS-TL) integrates the sulfide into the carbon and nitrogen-containing precursor $O$-acetylserine (OAS)[14]. The OAS formation limits the Cys biosynthesis rate and is catalyzed by the enzyme SERINE ACETYLTRANSFERASE (SERAT)[15]. OAS-TL and SERAT physically interact in the cysteine synthase complex (CSC). In contrast to other protein complexes composed of enzymes catalyzing consecutive steps in a biochemical pathway, the CSC does not facilitate substrate channeling, as OAS-TL is inactivated in the CSC but acts as a regulatory subunit that stimulates SERAT activity. The reversible and highly dynamic association of the CSC, and consequently SERAT activity, is controlled by sulfide and OAS, both of which display dual functions as the substrates of OAS-TLs and modulators of protein-protein interaction[15]. Sulfide stabilizes the CSC[16], whereas OAS dissociates it[17–19]. These biochemical properties define the CSC as a dynamic sensor of sulfur availability, controlling the net flux of cysteine biosynthesis. Higher plants possess CSCs in all subcellular compartments capable of translation, enabling them to fine-tune cysteine biosynthesis precisely to their demands[20,21]. The CSCs in the cytosol and mitochondria associate spontaneously. In contrast, the formation of the chloroplast-localized CSC (pCSC) is promoted by the *peptidyl-prolyl cis-trans* isomerase (PPlase) CYP20-3, which can bind 12-oxo-phytodienoic acid (OPDA), a precursor of the phytohormone jasmonic acid-Isoleucine (JA-Ile)[22,23]. This mechanism was proposed to promote cysteine biosynthesis for the production of the reactive oxygen species (ROS) scavenger glutathione during stress conditions, thereby maintaining the redox milieu of the stressed cell. JA-Ile is predominantly involved in the response to wounding and pathogen attack. Both stress conditions also require enhanced glutathione production to cope with the associated ROS formation. Remarkably, OPDA was also reported to specifically accumulate in response to drought stress in three Arabidopsis ecotypes, although JA was not induced under this stress[24]. In a more recent study, OPDA was found to act as a negative regulator of stomatal opening, directly connecting OPDA to stomatal aperture regulation[25]. Since sulfide, the downstream product of sulfate, and OPDA can trigger CSC formation, we supposed that the formation of CSCs might contribute to stomatal aperture regulation.

In this study, we report that the pCSC, but not the cytosolic CSC (cCSC) or the mitochondrial CSC (mCSC), acts as a sensor for abiotic stress and is indispensable for stomatal closure induced by water limitation and high light stress. We unveil that these two stress signal transduction pathways converge in the activation of the pCSC: (1) the xylem-delivered drought-stress signal sulfate results in sulfide provision for pCSC stabilization, and (2) OPDA-pCSC-ABA axis signals local high light stress via binding of high light-induced OPDA to CYP20-3, which promotes pCSC formation. This pCSC-dependent hub triggers ABA synthesis and signaling, defining the CYP20-3 pCSC complex as a stroma-localized OPDA sensor that connects oxylipin with ABA signaling, allowing for retrograde signaling routes in response to diverse stresses. In a gain-of-function approach, we engineered the protein-protein interaction of the OAS-TL B subunit of the pCSC to form a constitutively activated complex with SERAT2;1, resulting in increased cysteine synthesis and consequently more closed stomata, generating a drought-resistant phenotype. Since this approach shows no growth penalty in Arabidopsis and is easily engineered genetically, our findings provide a starting point for breeding crop plants with improved resistance to soil-drying and drought-associated high-light conditions.

## Results

### pCSC is essential for drought stress-induced stomatal closure
Sulfate acts as a long-distance signal of soil water limitation in diverse flowering plants, including the herbaceous crop maize and the tree poplar[7,9,26,27]. We recently demonstrated that sulfate must be incorporated into cysteine (Cys) to trigger ABA biosynthesis and stomatal closure in the model plant *Arabidopsis thaliana*[6] (Fig. 1a). However, Cys

can be synthesized in the cytosol, the chloroplasts/plastids, and the mitochondria in all so far analyzed vascular plants[14]. To investigate if Cys synthesis in a specific subcellular compartment is critical for ABA biosynthesis and stomatal closure, we tested the ability of guard cells in epidermal peels of plants lacking the CSC in the cytosol, the mitochondria or the plastids for sulfate-induced stomatal closure. Application of sulfate or ABA rapidly triggered stomatal closure in wild type and in mutants deficient in the cytosolic SERAT3 isoforms, the subunit SERAT or OAS-TL of the cytosolic CSC (cCSC) or the mitochondrial CSC (mCSC, Fig. 1b and Supplementary Fig. 1a, b). However, guard cells depleted of subunits (*SERAT2;1* and *OAS-TL B*) forming the CSC localized in the plastids (pCSC) failed to close stomata upon sulfate treatment, but could still respond to ABA (Fig. 1b and Supplementary Fig. 2a, b, d, e). Leaf-embedded stomata lacking the pCSC also did not respond to sulfate when fed via the xylem of the petiole, which is the native sulfate transport route during long-distance signaling of soil moisture (Fig. 1c). Remarkably, the other established chemical signal for soil water limitation, the peptide hormone CLE25, also failed to close isolated stomata or leaf-embedded stomata when the pCSC was absent (Fig. 1c and Supplementary Fig. 2c, f). We also tested the potential crosstalk between these two drought stress signals, sulfate and CLE25, and found no indication of sulfate-induced *CLE25* expression in roots and leaves, as well as CLE25-induced sulfate transporter (*Sultr*) genes in leaves (Supplementary Fig. 3a, b, d). However, in roots, several *Sultr* genes were slightly up- or down-regulated at the transcript level upon CLE25 treatment (Supplementary Fig. 3c), indicating a potential regulation of sulfur metabolism by CLE25 in this organ.

We previously used the established ABA signaling reporter *pRAB18::GFP*[28] to quantify endogenous ABA levels of wild type leaves in response to sulfate[6]. Since *RAB18* is transcriptionally induced in mesophyll and guard cells by ABA[29], we crossed *pRAB18::GFP* lines with mutants lacking the pCSC to quantify ABA accumulation in these mutants at cellular resolution. We found that in epidermal peels of the wild type, only guard cells responded to ABA or sulfate application with indistinguishable induction of GFP expression (Fig. 1d). In contrast, stomata lacking the pCSC responded to ABA but failed to respond to sulfate (Fig. 1d). In agreement with previous reports[8], the peptide hormone CLE25 induced ABA-responsive GFP expression in wild type stomata. However, stomata lacking the pCSC failed to trigger GFP accumulation in response to CLE25 (Fig. 1d). These results demonstrate that pCSC-triggered Cys biosynthesis is critical for the induction of ABA production by the drought stress signals, sulfate and CLE25, in guard cells.

Since both soil drying signals failed to induce ABA production and stomatal closure in pCSC mutants, we tested the sensitivity of mutants lacking the CSC in different compartments to soil drying. Exclusively, mutants deficient in the pCSC were sensitive to water withdrawal (Fig. 1e). Mutants lacking the cCSC or the mCSC were as tolerant as the wild type, albeit the mCSC contributes >85% to total net cysteine synthesis capacity in leaves[20]. Although neither mutant lacking only one subunit of pCSC (*serat2;1* and *oastlb*) has a lower cysteine content than wild type under normal conditions, both mutants are also soil-drying sensitive (Supplementary Fig. 4a, b), suggesting that pCSC formation is mandatory for the relevant induction of ABA synthesis in response to CLE25 or sulfate. Moreover, Soil drying led to a massive accumulation of foliar ABA, probably due to the induction of vascular ABA biosynthesis[30,31]. However, this massive accumulation occurred after soil drying-induced stomatal closure and was unaffected by the absence of pCSC (Supplementary Figs. 4c–e, and 5a–c), strongly suggesting that the pCSC-mediated ABA biosynthesis in guard cells probably is an initial signal to close the stomata and massive accumulation of ABA in leaves is rather a secondary signal to keep stomata closed. Furthermore, complementation of *oastlb* with At*OAS-TL B* restored wild type-like levels of OAS-TL B protein (Supplementary Fig. 6a), rescued the sensitivity of *oastlb* to soil drying, and enabled the

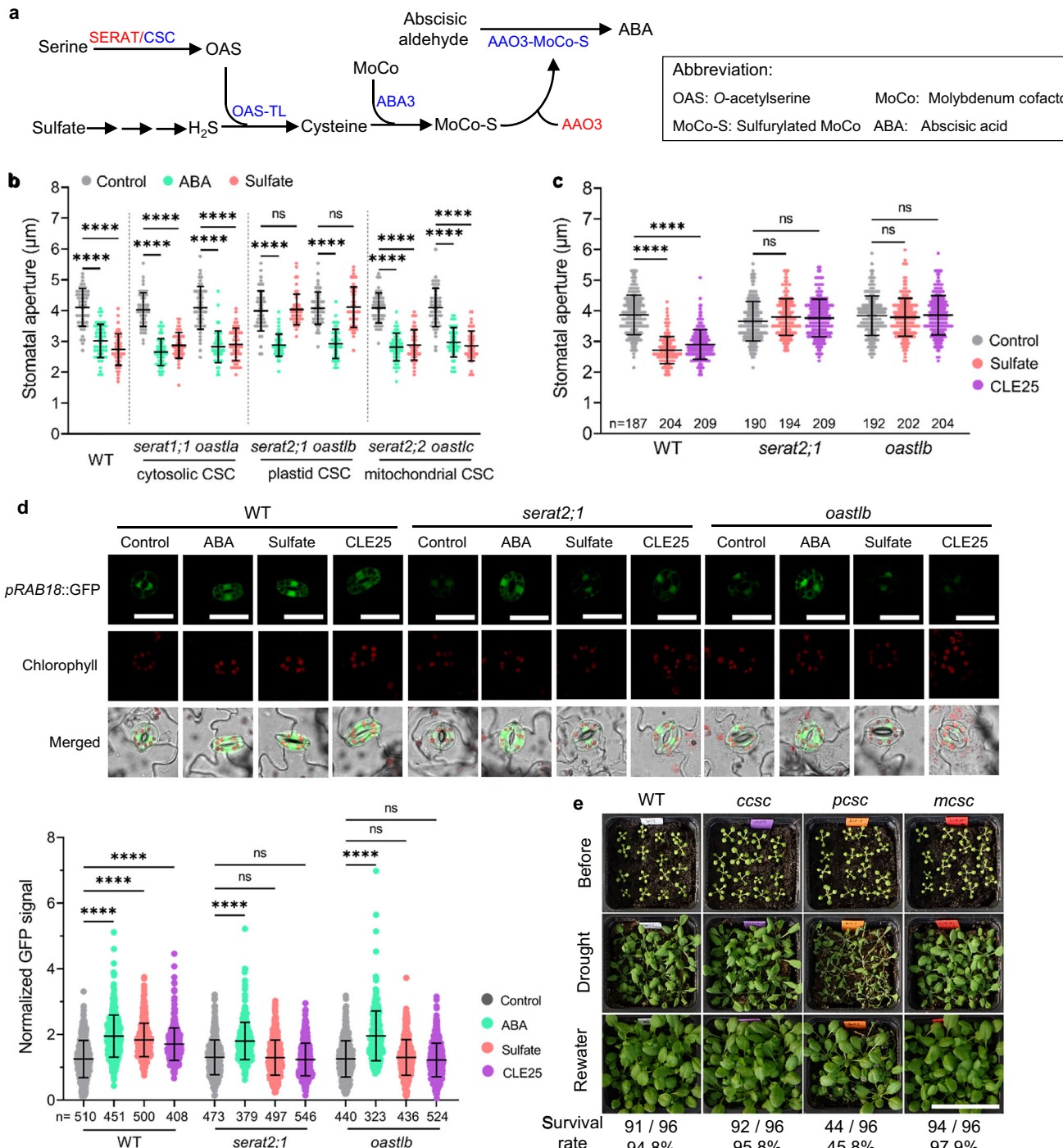

**Fig. 1 | Exclusively the pCSC is critical for soil-drying signals-induced ABA response and stomatal closure. a** Metabolic link between sulfur metabolism and ABA biosynthesis. Inactive and active enzymes are indicated in red and blue colors, respectively. Enzyme-catalyzed reactions are shown as arrows. SERAT limits cysteine synthesis and is activated by cysteine synthase complex (CSC) formation. **b** Stomatal apertures of wild type (WT, Col-0) and CSC mutants in different subcellular compartments. Isolated stomata in epidermal peels were treated with water (control), ABA, or sulfate for a defined time (*n* = 50). A representative result from three independent experiments is shown. **c** Apertures of imprinted stomata from detached leaves of WT and two pCSC mutants (*serat2;1* and *oastlb*) fed via the petiole with water (control), sulfate, or the peptide hormone CLE25 were analysed (*n* = 187–209). A representative result from at least two independent experiments is shown. **d** Impact of sulfate or CLE25 on ABA signaling in guard cells as revealed by determining the GFP fluorescence driven by the ABA reporter construct *pRAB18::GFP*. Representative

images of stomata from WT, *serat2;1*, or *oastlb* that were stably transformed with *pRAB18::GFP* and treated with water (control), ABA, sulfate, or CLE25. Scale bar, 20 μm. The lower left subpanel displays the quantification of GFP signals in guard cells from two independent experiments (*n* = 322–546). **e** Exclusively, mutants lacking both pCSC subunits (*pcsc*, *serat2;1oastlb* double mutant) are sensitive to soil-drying. 18-day-old seedlings of the WT and mutants lacking both subunits of the CSC located in the cytosol (*ccsc*, *serat1;1oastla* double mutant), the plastid (*pcsc*), and the mitochondria (*mcsc*, *serat2;2oastlc*) were subjected to water withdrawal for 15 days. Representative images were taken before drought (before), at the end of water withdrawal (drought), and 3 days after rewatering (rewater). The survival rate of genotypes is based on subjecting 96 individuals to water limitation. Scale bar, 5 cm. Data in **b**–**d** are shown as mean ± SD. Statistical differences were analyzed by one-way ANOVA followed by Tukey's test. \*\*\*\**P* < 0.0001. ns, no significant difference. Source data and *p* values are provided as a Source data file.

complemented OAS-TL B *oastlb* mutant to close its stomata in response to the soil-drying signals sulfate and CLE25 (Supplementary Fig. 6b–d). Taken together, our data uncover that the established xylem-delivered soil drying signals, sulfate and CLE25, merge at the pCSC for induction of ABA biosynthesis. Consequently, we name these previously undescribed signal transduction pathways Sulfate-pCSC-ABA and CLE25-pCSC-ABA.

## OPDA induces ABA-mediated stomatal closure via the pCSC

The phytohormone JA-Ile is a central regulator of diverse abiotic and biotic stress responses. Wounding triggers JA-Ile production and stomatal closure by activating COI1-dependent guard cell JA-Ile signaling[32]. OPDA is the precursor of the phytohormone JA-Ile, but possesses signaling roles distinct from JA-Ile[33,34]. Since OPDA accumulates specifically under drought stress in leaves and has been reported to support stomatal closure[24], we aimed to determine whether OPDA triggers stomatal closure via JA-Ile or an as-yet unknown mechanism that triggers ABA production. For this reason, we applied OPDA to mutants deficient in the production or sensing of JA-Ile or ABA. The conversion of OPDA to JA-Ile or perception of JA-Ile by the receptor COI1 was not required for OPDA-induced stomatal closure (Supplementary Fig. 7a–c). However, OPDA failed to trigger stomatal closure in the ABA biosynthesis mutants, *aba3-1* and *aao3-4*, or the ABA signaling mutants, *abi1-2* and *ost1-3* (Supplementary Fig. 7d, e). Moreover, like ABA, OPDA triggered a significant change in the FRET signal of the ABA sensor ABAleon2.1 in guard cells, demonstrating that OPDA stimulates ABA accumulation in guard cells (Supplementary Fig. 7f). These findings strongly suggest that OPDA-induced stomatal closure is achieved by regulation of ABA biosynthesis, rather than JA-Ile biosynthesis and JA-Ile signaling.

Since OPDA can promote pCSC formation[22,23], we tested whether OPDA triggers ABA production by stimulating Cys synthesis in plastids due to enhanced pCSC formation. OPDA triggered stomatal closure in the wild type and mutants lacking the cCSC, the mCSC or the cytosolic SERAT3 isoforms, (Fig. 2a and Supplementary Fig. 8). In contrast, pCSC mutants did not close stomata upon OPDA treatment for up to three hours, albeit the wild type closed the stomata after 1 h of OPDA application (Fig. 2a and Supplementary Fig. 9a, b). Moreover, ABA-responsive GFP expression by *pRAB18::GFP* was significantly induced by OPDA in wild type guard cells, but this induction was absent in guard cells of pCSC mutants (Fig. 2b). OPDA failed to decrease stomatal aperture when fed via the petiole (Supplementary Fig. 10a), suggesting that OPDA is a local stomatal closure signal. This finding is in agreement with the substantial hydrophobicity and high biochemical reactivity of the carbocyclic fatty acid, OPDA, that prohibit its efficient long-distance transport[35]. The plastid envelope membrane-originated OPDA facilitates pCSC formation by immediately binding to CYP20-3 in the stroma of plastids[22]. We therefore wanted to test whether CYP20-3 is critical for OPDA-induced ABA production, leading to stomata closure. Stomata in epidermal peels of two CYP20-3 deficient mutants failed to close in response to OPDA, but reacted like the wild type to ABA, sulfate, or CLE25 application with closure (Fig. 2c and Supplementary Fig. 10b). These findings uncover that CYP20-3 is explicitly required for OPDA-induced stomata closure, but does not contribute to the stimulation of ABA synthesis in response to CLE25 or sulfate. Furthermore, ABA-responsive GFP expression by *pRAB18::GFP* was also abolished in *cyp20-3* mutants after OPDA administration (Fig. 2d), strongly suggesting that the pCSC is a hub for integrating at least three stomatal closure signals by using specific co-receptors, like CYP20-3.

The OPDA conversion mutant, *opr3-1*, is deficient in the primary and peroxisome-localized OPDA reductase, OPR3, and suffers from closed stomata[24]. We confirmed the closed stomata phenotype of *opr3-1* and found that the *opr3-3* mutant also displays closed stomata (Fig. 2e and Supplementary Fig. 11a, b). Absence of cytosolic OPR2 had no

impact on stomata aperture (Supplementary Fig. 11a, b). The closed stomata phenotype of *opr3-1* was hitherto enigmatic and supposed to be caused by pleiotropic disturbance of the oxylipin pathway resulting in JA-Ile formation. Based on our findings, we reasoned that possibly OPDA accumulation in guard cells of *opr3* mutants triggers pCSC association, resulting in ABA-induced stomata closure. We therefore tested the stomata aperture in mutants deficient in OPDA transport from the plastids to the peroxisome, where OPDA is converted to JA (Supplementary Fig. 11a). The *jassy* mutant, lacking an OPDA exporter in the outer membrane of the chloroplast[36], was not affected in the stomata aperture (Supplementary Fig. 11b). However, the *jassy* mutant is not impaired in self-fertilization, implying that a backup OPDA transporting system exists at the outer membrane of plastids, enabling efficient OPDA export from the plastids into the cytosol under non-stressed conditions questioning substantial OPDA accumulation in this mutant. The ATP binding cassette (ABC) transporter PXA1, also known as CTS and PED3, is involved in peroxisomal OPDA import[37–39]. In contrast of *jassy* mutant, the *pxa1-1* mutant, which accumulates OPDA under non-stressed conditions[34,40], displayed constitutively closed stomata under non-stressed conditions (Supplementary Fig. 11c). In agreement with the hypothesis that OPDA-induced cysteine formation triggers ABA production, we found that *opr3-1* exhibits higher ABA-responsive GFP expression in guard cells, as well as significantly higher cysteine steady-state levels in *opr3-1* and *opr3-3*. However, neither mutant showed a statistically significant increase in OPDA or ABA steady-state levels in total leaves (Fig. 2f and Supplementary Fig. 11d, e). The connection between OPDA, cysteine, and ABA was also supported by the significant accumulation of OPDA and cysteine in ABA biosynthesis mutants (Supplementary Fig. 12a, b), suggesting the existence of negative regulatory feedback loops controlling OPDA and cysteine synthesis in response to ABA accumulation. Inhibition of JA-Ile sensing in the *opr3-1* mutant did not affect the stomatal closure phenotype of *opr3-1* (Supplementary Fig. 11f). In contrast, the crossing of ABA biosynthesis mutants, pCSC mutants, or *cyp20-3* into *opr3-1* re-opened the stomata as well as decreased ABA-responsive GFP expression in the guard cells in the respective double mutants and reversed the decreased water loss of *opr3-1* leaves (Fig. 2e, f and Supplementary Fig. 11g, h). The latter findings provide direct genetic evidence for the previously undescribed OPDA-pCSC-ABA signaling axis, as they demonstrate that endogenous OPDA accumulation in *opr3* only causes stomatal closure when CYP20-3 and a functional pCSC are present to stimulate ABA biosynthesis.

Since high light stress results in rapid OPDA accumulation and stomatal closure[2], we tested if short-term high light stress triggers stomatal closure in the wild type and mutants deficient in the pCSC, CYP20-3, or ABA biosynthesis. In contrast to the wild type, mutants lacking the OPDA-pCSC-ABA signaling axis did not close their stomata in response to mild and intense high light stress (Fig. 2g and Supplementary Figs. 13a–c and 14a). Moreover, we found that mutants defective in JA-Ile formation or JA-Ile perception were also impaired in this short-term response, as suggested by earlier studies[2] (Supplementary Fig. 14b, c). These discoveries position OPDA at the center of a signaling axis that converts high light stress into stomatal closure via the OPDA-pCSC-ABA signaling pathway.

## High light stress triggers pCSC-mediated ABA synthesis in guard cells

ABA is synthesized in all plant cells and effectively transported between them[41]. However, the ABA is sensed in the guard cells to trigger stomata closure. It is currently unclear which cell types are predominantly responsible for ABA synthesis in response to the diverse stresses leading to stomata closure. pCSC formation is limited by the subunit SERAT2;1[42], which is the CSC-forming SERAT isoform contributing least to total SERAT activity[20]. Based on the dominant expression of *SERAT2;1* in guard cells (Supplementary Fig. 15a–c), we

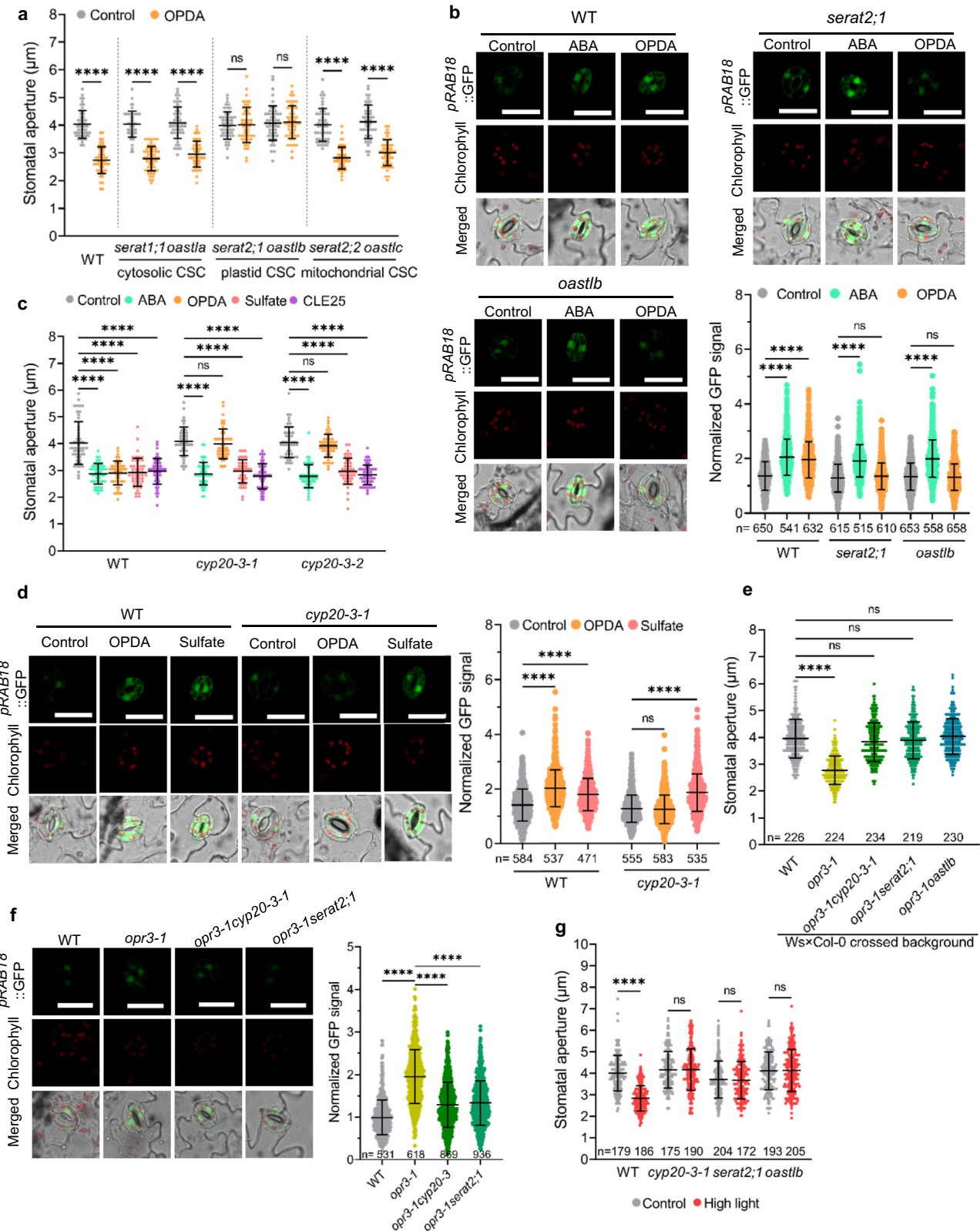

hypothesized that a substantial amount of ABA is synthesized in the guard cells after pCSC formation in response to soil drying or high light stress. To test this hypothesis, we applied two mutants lacking the guard cell-localized ABA importer ATP-binding cassette subfamily G member 40[43] (ABCG40) and analyzed their ability to respond to OPDA, sulfate, CLE25, and ABA. In contrast to stomata in epidermal peels of the wild type, stomata in epidermal peels of *abcg40-1* and *abcg40-2*

mutants remained open upon external ABA application (Fig. 3a and Supplementary Fig. 16). However, OPDA, sulfate, and CLE25 stimulated the closure of the *abcg40* stomata, demonstrating that these signals could trigger ABA production inside the guard cells (Fig. 3a). Even more importantly, leaf-embedded *abcg40* stomata responded to the xylem-delivered long-distance signals sulfate and CLE25 when fed via the petiole, but did not respond to xylem-delivered ABA (Fig. 3b).

**Fig. 2 | OPDA and high light triggers ABA production and stomatal closure via CYP20-3 and the pCSC. a** Apertures of stomata embedded in epidermal peels of WT and different CSC mutants treated with water (control) or OPDA (*n* = 50). **b** Impact of OPDA on ABA signaling in guard cells revealed by determining the GFP fluorescence driven by the ABA reporter construct *pRAB18::GFP*. Representative images of stomata from WT, *serat2;1*, or *oastlb* that were stably transformed with *pRAB18::GFP* and treated with water (control), ABA, or OPDA. Scale bar, 20 μm. The lower right subpanel displays the quantification of GFP signals in guard cells from three independent experiments (*n* = 515–658). **c** Apertures of stomata embedded in epidermal peels of WT and two *cyp20-3* mutants treated with water (control), ABA, sulfate, CLE25, or OPDA (*n* = 50). **d** Impact of OPDA and sulfate on ABA signaling in guard cells of WT and *cyp20-3-1*. Representative images of stomata from WT and *cyp20-3* stably transformed with *pRAB18::GFP* were treated with water (control), OPDA, or sulfate. Scale bar, 20 μm. The right subpanel displays the quantification of GFP signals in guard cells from two independent experiments (*n* = 471–584). **e** Apertures of imprinted stomata from leaves of WT, and *opr3-1* related mutants (*n* = 219–234). **f** ABA signaling in guard cells of WT, *opr3-1*, *opr3-1cyp20-3-1*, and *opr3-1serat2;1* plants. Representative images of stomata from WT, *opr3-1*, *opr3-1cyp20-3-1*, and *opr3-1serat2;1* plants stably transformed with *pRAB18::GFP*. Scale bar, 20 μm. The right subpanel displays the quantification of GFP signals in guard cells from three independent experiments (*n* = 531-936). **g** Apertures of imprinted stomata from leaves of WT(Col-0), *cyp20-3-1*, *serat2;1* and *oastlb* after high light treatment for 10 min (*n* = 172–205). Data in **a**, **c** are representative results from three independent experiments. Data in **e**, **g** are representative results from two independent experiments. Data in **a**–**g** are shown as mean ± SD. Data were analysed by one-way ANOVA followed by Tukey's test (**a**–**g**). ****$P < 0.0001$. ns, no significant difference. Source data and *p* values are provided as a Source data file.

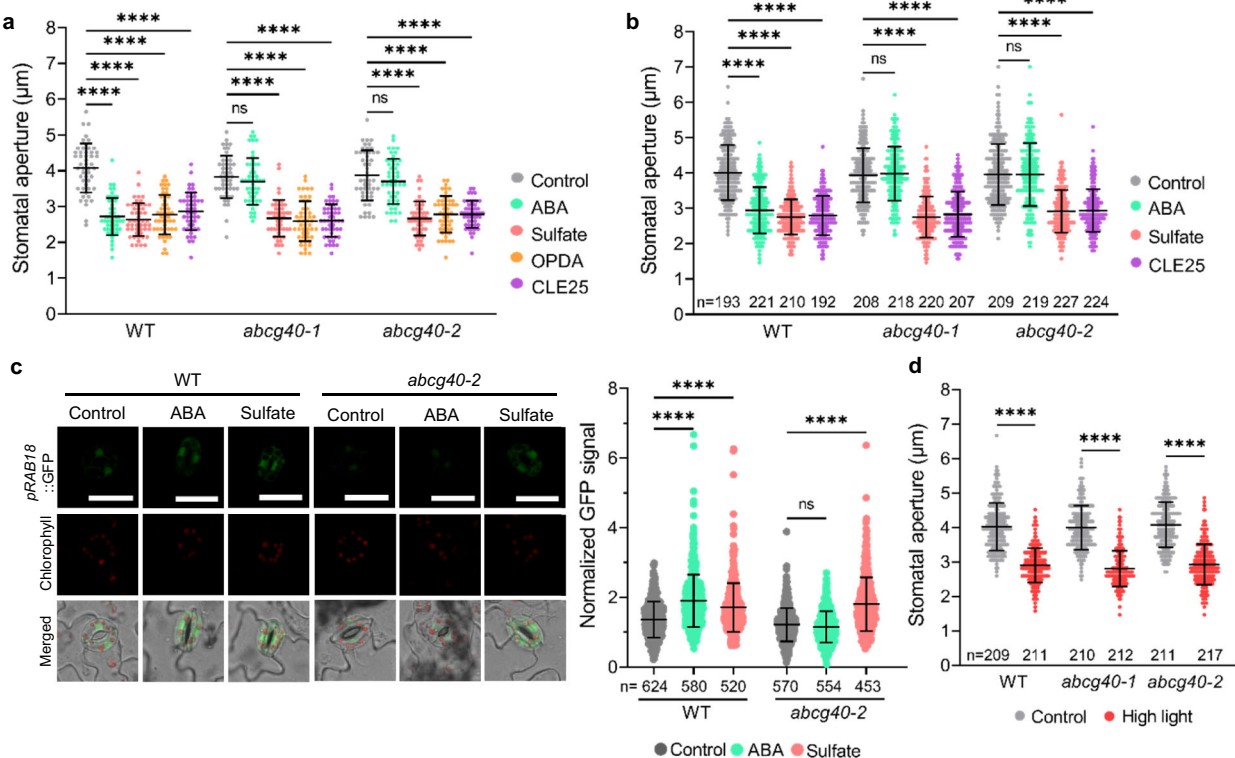

**Fig. 3 | Stress signals-induced ABA biosynthesis in guard cells is sufficient to trigger stomatal closure. a** Stomatal apertures of wild type (WT, Col-0) and two guard cell-localized ABA importer *abcg40* mutants. Isolated stomata in epidermal peels were treated with ABA, sulfate, CLE25, or OPDA (*n* = 50). Data are representative results from three independent experiments. **b** Apertures of imprinted stomata from detached leaves of WT and two *abcg40* mutants fed via the petiole with ABA, sulfate, or CLE25 (*n* = 192–227). **c** Impact of ABA or sulfate on ABA signaling in guard cells as revealed by determining the GFP fluorescence driven by the ABA reporter construct *pRAB18::GFP*. Representative images of stomata from WT or *abcg40-2* that were stably transformed with *pRAB18::GFP* and treated with water (control), ABA, or sulfate. Scale bar, 20 μm. The right subpanel displays the quantification of GFP signals in guard cells from three independent experiments (*n* = 453–624). **d** Apertures of imprinted stomata from leaves of WT and two *abcg40* mutants after high light treatment for 10 min (*n* = 209–217). Data in **a**–**d** are shown as mean ± SD. Data were analysed by one-way ANOVA followed by Tukey's test. ****$P < 0.0001$. ns no significant difference. Source data and *p* values are provided as a Source data file.

Furthermore, we crossed the ABA reporter construct *pRAB18::GFP* into the *abcg40-2* mutant and found that stomata lacking the ABCG40 failed to respond to ABA but responded to sulfate (Fig. 3c). These findings strongly suggest that pCSC-triggered ABA production occurs predominantly in guard cells, which is in agreement with the notion that CLE25 and sulfate are root-originated and xylem-delivered soil drying signals that trigger ABA production in leaves[7,8].

Since OPDA is induced upon high light stress in leaves[2] and can trigger ABA biosynthesis via pCSC formation (previous section), we applied short-term high light stress to leaves of *abcg40-1* and *abcg40-2* to test if ABA transport into the guard cells is required for immediate stomata closure upon this stress. Both mutants closed the stomata

with the same kinetic as the wild type (Fig. 3d), demonstrating that ABA transport via ABCG40 is irrelevant for immediate high light-induced stomata closure. Since the described stress conditions trigger short-term responses, these findings do not exclude an important function of ABCG40 in maintaining stomata closed under prolonged stress (see also discussion).

Next, we applied the guard cell-specific complemented ABA biosynthesis mutant *MYB60::ABA3 aba3-1*[44], to test the importance of guard cell-autonomous ABA biosynthesis for OPDA-triggered stomatal closure. In contrast to *aba3-1* (Supplementary Fig. 7d), the *MYB60::ABA3 aba3-1* closed the stomata in response to OPDA like the wild type (Supplementary Fig. 17). These results strongly suggest that

pCSC-based guard cell-autonomous ABA biosynthesis is sufficient for high light stress-triggered stomatal closure. We cannot entirely exclude that other cell types contribute cysteine for cysteine-triggered ABA production in guard cells since the pCSC is also present in mesophyll cells (Supplementary Fig. 15b). However, such a hypothetical contribution of cysteine, which is generated in other tissues, appears unlikely, given the prompt response (<1.5 min) of stomata to high light[2] and the distance of extracellular transport between the plasmodesmata-lacking guard cells and the vascular/mesophyll cells. In addition, the pavement cells that surround the guard cells are depleted of chloroplasts in Arabidopsis and thus lack significant OPDA production or the pCSC.

### Constitutive pCSC association causes drought tolerance

Intracellular cues tightly control the reversible association of subunits into CSCs in the cytosol, the plastids, and the mitochondria. While sulfate-originated sulfide and light-induced OPDA promote CSC formation[16,22], the cysteine precursor OAS dissociates the CSC by outcompeting the C-terminus of SERAT from the active site of OAS-TL[45]. CSC dissociation by OAS is a critical negative feedback loop, allowing plants to adjust cysteine synthesis flux in response to sulfur deficiency and other stresses[14]. Based on distinct structure-function relationships in OAS-TL[17,46], we genetically engineered two OAS-TL B protein variants that form the pCSC but differ in their CSC dissociation response to OAS. OAS-TL B(S172N) allowed the dissociation of the pCSC by OAS, while OAS-TL B(M167A) formed a stable CSC that was resistant to OAS dissociation (Fig. 4a and Supplementary Fig. 18a, b). Both OAS-TL B variants were enzymatically inactivated to ensure they only serve as regulatory subunits in the pCSC (Fig. 4b). Ectopic expression of the inactivated OAS-TL B(M167A) driven by the constitutive 35S-promotor resulted in massive OAS-TLB mRNA accumulation. However, it did not increase total OAS-TL activity in leaves or impair plant growth and seed yields under non-stressed conditions (Supplementary Fig. 19a–h). We previously demonstrated that constitutive pCSC formation substantially enhances the cysteine synthesis flux in rice[17]. As expected, only Arabidopsis plants expressing OAS-TL B(M167A) accumulated OAS and cysteine as a consequence of enhanced SERAT activity in the CSC, while OAS-TL B(S172N) overexpressors displayed wild type-like OAS and cysteine levels (Supplementary Fig. 19i). Constitutive pCSC formation caused permanently decreased stomata aperture and slightly increased accumulation of ABA in OAS-TL B(M167A) leaves (Fig. 4c, d). In contrast, plants expressing OAS-TL B(S172N) displayed wild type-like stomata aperture and wild type-like ABA steady state levels (Fig. 4c–d). The partially closed stomata resulted in lower stomatal conductance in OAS-TL B(M167A) plants (Fig. 4e), which explains the decreased water loss of detached OAS-TL B(M167A) leaves when compared to wild type (Fig. 4f). As expected, OAS-TL B(M167A) expressing plants accumulated ABA in the cytosol of guard cells as indicated by significantly higher ABA-responsive GFP expression in the three independent OAS-TL B(M167A) lines (Fig. 4g). As most important physiological consequence these lines showed enhanced resistance to soil water deficit (Fig. 4h). In summary, the constitutively enhanced ABA biosynthesis in stomata of OAS-TL B(M167A) expressing plant caused partial stomatal closure and resulted in significant higher tolerance to soil drying.

Next, we introduced the OAS-TL B(M167A) mutant protein into *serat2;1* and *oastlb* to provide further evidence for the hypothesis that the constitutive stomatal closure phenotype of stable pCSC mutants is caused by enhanced cysteine formation in plastids. The crossing with *serat2;1* inhibited the accumulation of OAS and cysteine in the double mutants (Fig. 5a), since SERAT2;1 could not be activated by stable pCSC formation. The OAS-TL B(M167A) *oastlb* double mutants could still activate SERAT2;1 but cannot convert the OAS to cysteine in the chloroplast. Consequently, these double mutants still accumulated OAS in plastids and cysteine, probably produced in the cytosol[21] from plastid-generated OAS (Fig. 5b). Remarkably, inhibition of plastid cysteine synthesis capacity re-opened the stomata in OAS-TL B(M167A) *serat2;1* and OAS-TL B(M167A) *oastlb* double mutants (Fig. 5c), and also reverted the decreased water loss phenotype of OAS-TL B(M167A) expressing plants (Fig. 5d). To demonstrate that the enhanced cysteine biosynthesis in plastids triggers stomatal closure by stimulating ABA biosynthesis, we introduced OAS-TL B(M167A) in mutants defective in ABA biosynthesis (*aba3-1*). Inhibition of ABA biosynthesis in OAS-TL B(M167A) expressing *aba3-1* lines did not impair OAS and cysteine accumulation (Supplementary Fig. 20a). However, it caused the re-opening of stomata (Supplementary Fig. 20b), which correlated with significantly lower foliar ABA levels in the double mutant compared to the OAS-TL B(M167A) lines (Supplementary Fig. 20c). Taken together, these findings suggest that constitutively activated pCSC-induced stomatal closure is dependent on plastid cysteine biosynthesis and ABA biosynthesis.

## Discussion

ABA is an essential stress hormone that kicks in when plants face tough environmental challenges like drought or intense sunlight. When stress hits, ABA steps up to close the stomata on leaves, reducing water loss and slowing down carbon dioxide intake to help plants conserve resources and survive[47,48]. In this study, we reveal that the pCSC acts as a central control hub, taking in various stress signals—including sulfate, CLE25, and OPDA—to boost ABA production for stomata closure. This physiological strategy fine-tunes stomatal function to optimize water efficiency and respond to at least two agronomically highly relevant environmental stresses, drought and high light (see Fig. 6).

Albeit most of the CSC-related net cysteine synthesis capacity is present in mitochondria[20], only the pCSC plays a pivotal role in driving ABA biosynthesis and stomatal closure in response to stresses mediated by sulfate or OPDA accumulation (Figs. 1b, 2a). This finding is consistent with the previous reports on sulfate-triggered stomatal closure, demonstrating that sulfate must be transported into the plastids and reduced to sulfide to close stomata[6,12]. Since sulfide and OPDA trigger pCSC formation, we conclude that pCSC stabilization boosts SERAT activity to fuel cysteine biosynthesis and ultimately trigger ABA production under harsh conditions. CLE25, on the other hand, is detected by BARELY ANY MERISTEM (BAM) receptors on the cell membrane[8]. Our findings suggest that pCSC-based cysteine biosynthesis acts downstream of CLE25 to drive ABA production and close stomata (Fig. 1b, c), though further study is needed to map the CLE25 and pCSC connection.

We discovered that short-term high light stress-induced stomatal closure is transduced via the OPDA-pCSC-ABA axis (Fig. 2g and Supplementary Figs. 13a–c, 14a). A recent report offers a mechanistic explanation for the high light-induced pCSC formation. In this scenario, OPDA binding to CYP20-3 enables the transfer of electrons from thioredoxin F2, an electron carrier in the photosynthesis reaction, to SERAT2;1, which splits the dimeric SERAT2;1 trimers in half to enable the formation of the pCSC. pCSC formation and its metabolic product, glutathione, were then shown to coordinate the expression of OPDA-responsive genes, optimizing plant survival under high light stress and other stresses that impinge on the production of reactive oxygen species[49]. These findings suggest that OPDA-triggered pCSC formation controls transcriptional responses not only by facilitating ABA production but also by promoting the synthesis of other cysteine downstream metabolites, such as glutathione. However, the OPDA-pCSC-ABA signaling axis may also participate in the control of stomatal movement driven by other stimuli, e.g., OPDA-triggered ABA synthesis may explain how OPDA inhibits stomata opening in response to blue light[25]. Remarkably, OPDA itself accumulates in the leaves of three tested Arabidopsis ecotypes under soil-drying conditions[24], whereas JA does not, aligning with OPDA's role in prompting stomatal closure through ABA rather than JA (Supplementary Fig. 7). Since the three

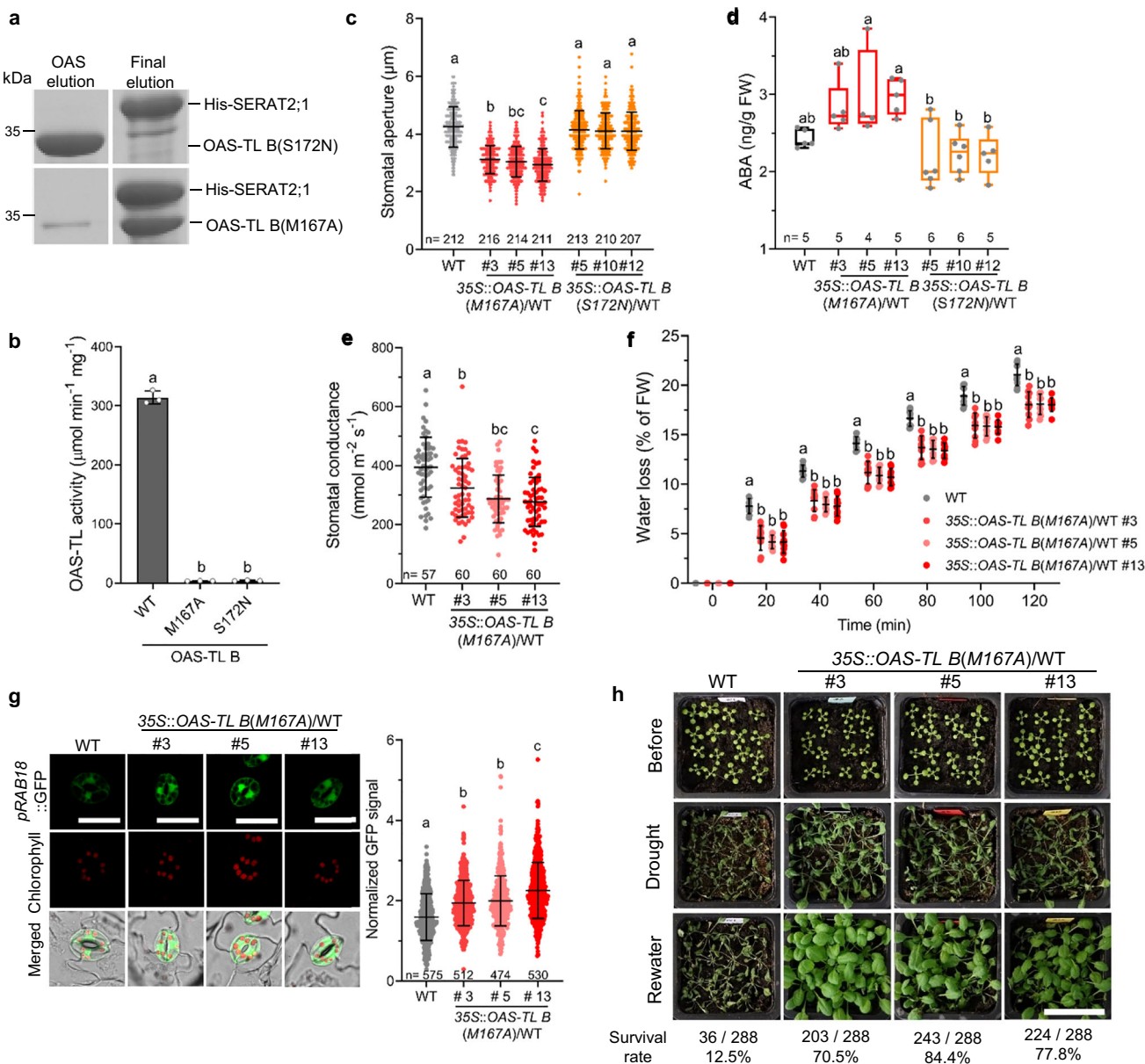

**Fig. 4 | Genetic engineering of a constitutively activated pCSC causes drought tolerance in Arabidopsis. a** In vitro analysis of the OAS dissociation capability of pCSC. OAS dissociates the pCSC formed by His-SERAT2;1 and OAS-TL B(S172N) whereas the pCSC including OAS-TL B(M167A) is resistant to OAS dissociation. **b** In vitro OAS-TL enzyme activity of OAS-TL B proteins ($n = 3$ technical replicates). **c** Apertures of imprinted stomata from leaves of 7-weeks-old wild type (WT, Col-0), *35S::OAS-TL B(M167A)* and *35S::OAS-TL B(S172N)* plants ($n = 207$-216). A representative result from three independent experiments is shown. **d** ABA concentrations of the leaves of 7-weeks-old WT, *35S::OAS-TL B(M167A)* plants and *35S::OAS-TL B(S172N)* plants ($n = 4$–6 biological replicates). **e** Stomatal conductance of WT and *35S::OAS-TL B(M167A)* plants. Data from four independent experiments are shown ($n = 57$–60). **f** Water loss of WT and *35S::OAS-TL B(M167A)* plants. Data from two independent experiments are shown ($n = 10$). **g** ABA signaling in guard cells of WT and *35S::OAS-TL B(M167A)* plants as revealed by determining the GFP fluorescence driven by the ABA reporter *pRAB18::GFP*. Representative images of stomata from

WT, and *35S::OAS-TL B(M167A)* plants stably transformed with *pRAB18::GFP*. Scale bar, 20 µm. The right subpanel displays the quantification of GFP signals in guard cells from two independent experiments ($n = 474$–575). **h** *35S::OAS-TL B(M167A)* plants are tolerant to soil-drying. 18-day-old seedlings were subjected to drought stress by withholding water for 17 days. Images were taken before drought (before), 17 days after drought (drought) and 3 days after rewater plants (rewater). The survival rate of genotypes is based on subjecting 288 individuals to water limitation in three independent experiments. Scale bar, 5 cm. Data in **b**–**c**, **e**–**g** are shown as means ± SD. Data in **d** are shown as boxplot. The box plots display medians (horizontal lines), 25% to 75% ranges (boxes) and min to max (whiskers). Data were analysed by one-way ANOVA followed by Tukey's test (**b**–**e**, **g**), or two-way ANOVA followed by Tukey's test (**f**). Different letters indicate significant differences among different genotypes ($P < 0.05$). Source data and $p$ values are provided as a Source data file.

---

ecotypes piled up OPDA and JA upon wounding, Arabidopsis tightly controls the conversion of OPDA to JA upon specific stresses[24]. In stomata possessing plants, the COI1 protein senses JA after conjugation to isoleucine as JA-Ile. However, in bryophytes, COI1 senses dn-OPDA[50], suggesting that during the evolution of the vessels and

improved water management by stomata, OPDA achieved novel functions and was replaced by JA-Ile, which serves in flowering plants as a specific signal for wounding and biotic stresses[51].

ABA synthesis directly within guard cells helps them react quickly to changes in humidity[44], making this process vital for survival in arid

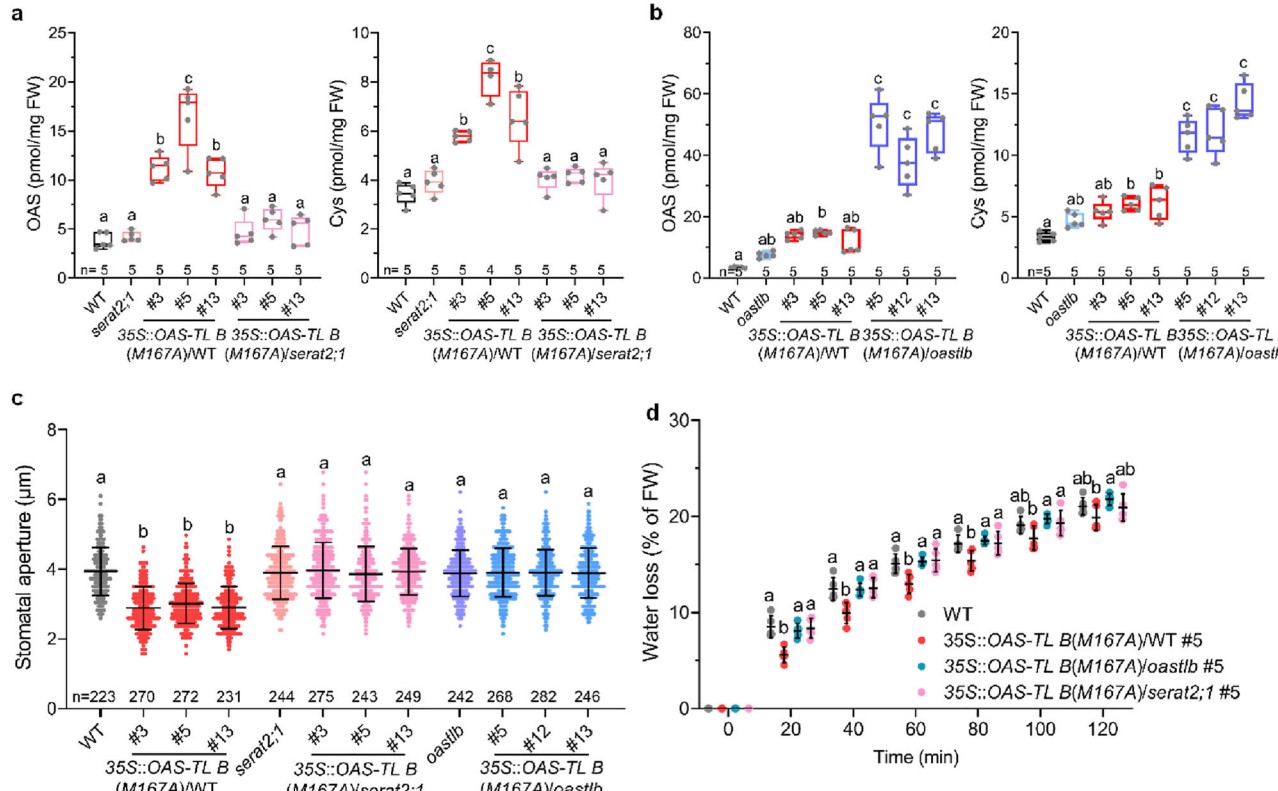

**Fig. 5 | Constitutively activated pCSC-induced stomatal closure is dependent on plastid cysteine biosynthesis. a, b** OAS and Cys concentrations in the leaves of wild type (WT), *serat2;1, oastlb*, or plants expressing OAS-TLB(M167A) in the WT, *serat2;1* (**a**) or *oastlb* (**b**) background. Data in **a** and **b** are shown as boxplots ($n = 4–5$ biological replicates). The boxplots display medians (horizontal lines), 25% to 75% ranges (boxes), and the minimum and maximum values (whiskers). **c** Apertures of imprinted stomata from detached leaves of WT and *35S::OAS-TL B(M167A)* plants in WT, *serat2;1* or *oastlb* background ($n = 223–282$). A representative result of two independent experiments is shown as mean ± SD. **d** Water loss of WT and plants expressing *OAS-TL B(M167A)* in the WT, *serat2;1* or *oastlb*. Data are shown as mean ± SD ($n = 5$). Data were analysed by one-way ANOVA followed by Tukey's test (**a**–**c**) or two-way ANOVA followed by Tukey's test (**d**). Different letters indicate significant differences among different genotypes ($P < 0.05$). Source data and $p$ values are provided as a Source data file.

conditions[52]. Our study indicates that early stress signals like sulfate, CLE25, and OPDA all contribute to cysteine production in guard cells (Fig. 3a), spurring ABA-driven stomatal closure when sunlight is harsh, or soil moisture is low. Supporting evidence also comes from isotope-labeled ABA tracer experiments in mutants of ABCG25, which mediates root-to-shoot translocation of ABA and ABA-glucosyl ester. Although the *abcg25* mutant was impaired in root-to-shoot ABA translocation, foliar ABA levels and stomatal closure during drought were unaffected in *abcg25*[53]. However, these findings do not contradict the crucial role of vascular ABA biosynthesis and ABA transport via ABCG40 into the guard cells in later stages of drought stress[41,54]. During these later stages of drought stress, the $CO_2$ concentration in the mesophyll decreases, generating stomata opening signals that must be counterbalanced. ABA biosynthesis in mesophyll cells and vascular cells contributes to the counterbalance of these signals, integrating the water supply status of these tissues into the water-limitation response of the stomata. Thus, the identification of guard cell specific ABA triggered by the pCSC formation does not contradict the important roles of ABA transporter during the drought stress response. In this scenario, ABA synthesis in guard cells is critical for early closure in response to environmental stimuli and import from leaves and root-generated ABA is essential for integrating the water supply status of cell in later stages of drought.

Interestingly, while ABA helps plants manage stress, it also acts as a growth-inhibiting hormone, often leading to early leaf aging across various species[3,55]. To handle this delicate balance, plants have developed precise mechanisms to fine-tune ABA levels in response to environmental pressures. We found that ectopic expression of the modified OAS-TL B(M167A) protein resulted in only non-harmful ABA accumulation in leaves. However, the OAS-TL B(M167A) expressing plants benefited from the stimulated ABA production predominantly in guard cells with substantially enhanced drought tolerance (Fig. 4c–h and Supplementary Fig. 19c–f). Since sulfate functions as a conserved drought signal across plant species[7,9,26,27], constantly activating the pCSC may offer a promising approach to genetically engineering drought-resistant crops.

In summary, our findings uncover the pCSC as a central hub for stress-induced regulation of ABA biosynthesis in guard cells and stomatal closure response, which is crucial for soil water limitation and high light stress. Whether this regulatory hub is also critical for ABA regulation in other cell types remains enigmatic due to the dominant expression of the pCSC subunit SERAT2;1 in guard cells and the specific adaptations of sulfate assimilation in this cell type[44].

## Methods

### Plant materials and growth conditions

*Arabidopsis thaliana* ecotype Columbia (Col-0) was used as the genetic background of the transgenic lines and as a wild-type control for almost all of the experiments, except for *opr3-1* mutant, which was in the Wassilewskija (Ws) background. The cysteine synthase complex mutants (*serat1;1, serat2;1, serat2;2, oastla, oastlb* and *oastlc*) were generated in earlier studies[20,21]. Two alleles of OPDA receptor *cyp20-3* mutants are described in ref. [56]. The ABA biosynthesis related mutants (*aba3-1, nced3-2*) were described previously[6]. The *aao3-4, ost1-3, abcg40-1*, and *abcg40-2* were obtained from the European Arabidopsis Stock Center[43,57,58]. *abi1-2*[59] was obtained from Prof. Rainer Hederich

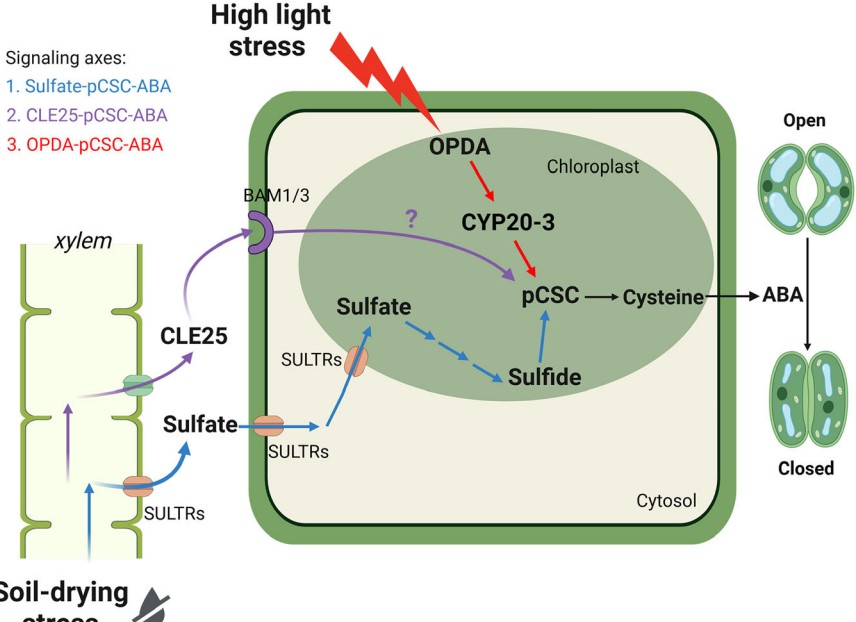

**Fig. 6 | Schematic diagram highlighting the critical role of pCSC in integrating drought and high light stress via the sulfate-pCSC-ABA, the CLE25-pCSC-ABA, and the OPDA-pCSC-ABA axes.** Drought stress triggers the transport of long-distance stress signals, such as sulfate and the peptide hormone CLE25, from the roots to the shoots to adjust water use efficiency. The CLE25 peptide binds at the cell surface to its receptor and induces stomatal closure in a pCSC-dependent manner. However, the interaction between CLE25 signaling and the pCSC remains enigmatic. The other drought-stress signal, sulfate, is taken up by leaf cells and reduced to sulfide in the chloroplast to trigger the stabilization of the pCSC. High light stress causes rapid accumulation of OPDA in the chloroplasts of leaf cells. There, OPDA stimulates pCSC formation after specific binding to the co-receptor CYP20-3. The stress-induced pCSC formation results in the production of cysteine, which serves as the substrate for the molybdenum cofactor sulfurylase ABA3. The production of sulfurylated MoCo is essential for the activation of AAO3, which synthesizes ABA. ABA triggers stomatal closure by the canonical ABA signaling pathway. Schematic created in BioRender. Wirtz, M. (2025) https://BioRender.com/tstk3cc.

(University of Würzburg, Germany). The JA-related mutants *aos*[60], *opr3-1*[61], *coi1-t*[62] were obtained from Dr. Ellen Hornung (Georg-August-Universität Göttingen, Germany), *coi1-16*[63] was a kind gift of Prof. Roberto Solano (National Center for Biotechnolgy, Spain) and *opr2-1*[64], *opr3-3*[64], *jassy*[36], *pxa1-1*[39], and *jar1-11*[65] were received from the European Arabidopsis Stock Center.

Plants were grown on soil under short-day conditions (8.5 h light, 100 μmol m$^{-2}$ s$^{-1}$ at 22 °C and 15.5 h dark at 18 °C) for all experiments. For transformation of plants and seed production, 10-week-old plants grown under shot-day conditions were transferred to long-day conditions (16 h light, 100 μmol m$^{-2}$ s$^{-1}$ at 22 °C and 8 h dark at 18 °C). For screening of positive transgenic plants, seeds were sterilized in 8% sodium hypochlorite solution for 15 min, washed four times with water, and then sowed on ½ Murashige and Skoog (MS) medium, which was supplemented with 50 μg ml$^{-1}$ hygromycin B (Sigma-Aldrich) and solidified with 0.6 % agar. The seedlings were stratified at 4 °C for 3 days in the dark and then grown for 2 weeks under short-day conditions in a growth chamber. After selection, the positive seedlings were transferred to soil. Plants deficient in JA-Ile production (*aos*, *opr3-1*, *opr3-3* and plants crossed with *opr3-1*) are infertile due to retarded anthers growth and were sprayed with 0.1 % MeJA (Sigma-Aldrich) in 0.1% Tween 20 to obtain seeds.

### Plasmid construction and transgenic plants
The OAS-TL B mutants (M167A and S172N) were generated by site-directed mutagenesis using primers defined in Supplemental Table 1. For functional assays in *E. coli*, the signal peptides in the N terminus of SERAT2;1 and OAS-TL B proteins were deleted before expression in *E. coli* (40 and 60 amino acid residues, respectively). For in vitro enzyme assays, gene fragments of *OAS-TL B*, *OAS-TL B(M167A)* and *OAS-TL B(S172N)* were cloned into the NcoI/BamHI sites of pET30a to generate Histone (His)-tag fused proteins, respectively. For in vitro pull-down

assays, mature *SERAT2;1* was cloned into the NcoI/BamHI sites of pET30a to generate a His-SERAT2;1 fusion protein, and gene fragments of *OAS-TL B(M167A)* and *OAS-TL B(S172N)* were cloned into the NcoI/BamHI sites of pET3d to generate untagged proteins, respectively.

For the complementation of *oastlB*, and the generation of plants expressing mutagenized OAS-TL B variants, gene fragments of *OAS-TL B* variants were cloned into pMDC32 vector using the Gateway technology. In brief, the open reading frames (ORFs) of *OAS-TL B* and *OAS-TL B(S172N)* were PCR-amplified using *att*B adapter-containing primers (Supplemental Table 1), and cloned into pDONR vector using BP Clonase (Invitrogen), respectively. *OAS-TL B-pDONR* and *OAS-TL B(S172N)-pDONR* were subcloned into pMDC32 vector using LR Clonase (Invitrogen), respectively. OAS-TL B(M167A) were PCR amplified and cloned into the AscI/PacI sites of pMDC32-35S-OAS-TL B(S172N) plasmid by replacing OAS-TL B(S172N) fragment. The respective primers are defined in Supplementary Table 1. The Arabidopsis transgenic plants 35S::OAS-TL B(S172N) in wild type background (#5, #10, #12), 35S::OAS-TL B(M167A) in wild type background (#3, #5, #13), 35S::OAS-TL B(M167A) in *oastlb* background (#12), and 35S::OAS-TL B(M167A) in *aba3-1* background (#1, #2, #6), were generated by *Agrobacterium*-mediated floral dip transformation. The Arabidopsis transgenic plants 35S::OAS-TL B(M167A) in *serat2;1* background (#3, #5, #13), 35S::OAS-TL B(M167A) in *oastlb* background (#5, #13), and 35S::OAS-TL B(M167A) in *abcg40-2* background (#13), were generated by crossing diverse mutants with plants expressing 35S::OAS-TL B(M167A) in wild type background (#3, #5, #13), respectively.

### Stomatal aperture bioassay
Epidermal peels were obtained from the abaxial side of Arabidopsis leaves (derived from at least 5 individual plants) and treated with different effectors as described previously[6], with the following modifications. Peels from leaves of at least 4 individual plants were floated

on stomata opening buffer[9] (50 mM KCl, 10 mM MES, pH 5.5) for 2 h under constant light and washed in distilled water (pH 5.5) for 5 min. Subsequently, the peels were transferred to distilled water (pH 5.5) supplemented without (control) or with different effectors for different times (15 mM $MgSO_4$ for 3 h, 50 μM ABA (Sigma-Aldrich) for 1 h, 10 μM OPDA (Cayman chemical) for 2 h, 1 μM CLE25 for 3 h). In addition, incubation time and concentrations of effectors used in time-course and dose-effect experiments were applied as indicated in respective figure legends. CLE25 peptide (RKVPNGPDPIHN) was synthesized from GENECUST (France). Stomata were imaged with the microscope (Leica DMIRB) and the width of the stomatal aperture was determined with ImageJ (version 1.52a)[66]. Each experiment was performed at least twice by two researchers and showed comparable results. Images were always analyzed in a double-blinded manner to avoid any bias during the analysis.

For petiole feeding experiments, plant leaves (derived from at least 4 individual plants) were detached by cutting the petioles and placing them in 2 ml Eppendorf tubes containing distilled water (pH 5.5) for 2 h. Subsequently, the detached leaves were transferred to 2 ml Eppendorf tubes containing distilled water (pH 5.5, as control) or distilled water (pH 5.5) supplemented with different effectors (15 mM $MgSO_4$, 50 μM ABA, 10 μM OPDA, 1 μM CLE25) and incubated for 3 h under constant light conditions.

For high light stress experiments, local leaves of plants were exposed to different high light intensity (300, 500, 1000, 2000 μmol $m^{-2} s^{-1}$) at 22 °C for 10 min using a halogen cold light source (PL3000, Photonic) as described previously[2].

For the determination of the stomatal aperture in untreated leaves or leaves subjected to petiole feeding or high light stress application, the abaxial side of leaves were imprinted with superglue liquid (UHU) and stomata were imaged with the microscope (Leica DMIRB) prior calculation of stomata aperture with ImageJ (version 1.52a)[66]. Stomata imaging and measurement were performed in a double-blinded manner to avoid any bias during the analysis.

## Metabolites analysis

To measure OAS and non-protein thiols (Cys and GSH), metabolites were extracted from 50 mg plant tissue with 300 μl of 0.1 M HCl. After centrifugation, OAS and thiols in the supernatant were labeled with AccQ-tag and monobromobimane (mBB), respectively. The labeled metabolites were quantified by reverse-phase ultra-performance liquid chromatography (UPLC, Waters) with an external standard curve as described previously[67].

For measurement of OPDA and ABA, phytohormones were extracted with methyl tert-butyl ether (MTBE) solution according to Salem et al.[68] and determinated by UPLC-MS/MS. In brief, 40 mg frozen plant material was extracted with 750 μl MTBE for 30 min at 4 °C. After centrifugation, the supernatants were mixed with an equal volume of 0.1% HCl and incubated for another 30 min at 4 °C. After centrifugation, the upper phase (green color) was transferred to a new 1.5 ml tube and dried in a vacuum concentration system (SpeedVac, Eppendorf). The dried samples were dissolved in 50 μl 50% MeOH. Phytohormones were separated by UPLC (Waters) and detected by QTRAP 6500+ mass spectrometry system (Sciex) and quantified with standard curve.

## RNA extraction and quantitative real-time PCR

Total RNA was extracted from plant tissues using a Universal RNA purification kit (EURx) following the manufacturer's protocol. First-strand cDNA was synthesized from RNA using FastGene Scriptase II cDNA Kit (Nippon Genetics) according to the manufacturer's instructions. Quantitative real-time PCR was performed on a Rotor-Gene Q cycler (Qiagen) using a qPCRBIO SyGreen Mix Lo-ROX (PCR Biosystems) according to the manufacturer's instructions. The Arabidopsis PP2A gene (AT1G13320) was used as the internal reference. The expression level of each gene was calculated as $2^{-\Delta\Delta Ct}$ relative to the internal reference. The primers used for qRT-PCR are listed in Supplementary Table 1.

## Plant protein extraction and immunoblotting analysis

Total soluble proteins were extracted from grinded leaves (0.5 g) with 500 μl protein extraction buffer (50 mM Hepes / KOH pH 7.4, 10 mM KCl, 1 mM EDTA, 1 mM EGTA, 10 mM DTT) supplemented with 1× protease inhibitor cocktail (Roche). Proteins were denatured by heating at 95 °C for 10 min and analyzed by immunoblotting. The proteins were detected by a rabbit anti-OAS-TL C antibody (against full-length Arabidopsis OAS-TL C, 1:2000 dilution) and HPR-conjugated goat anti-rabbit IgG (AS10852, Agrisera, 1:20000 dilution) was used as the secondary antibody. HPR activity was detected with the WesternBright Chemilumineszenz Substrat (Biozym), and quantified using an ImageQuant LAS 4000 (GE Healthcare).

## Determination of OAS-TL enzyme activity

Enzymatic activity of recombinant OAS-TL B protein or OAS-TLs in leaf extracts of plants were measured respectively, as described previously[17,21]. In brief, expressions of recombinant His-OAS-TL B proteins were induced with 1 mM isopropyl-β-D-thiogalactopyranoside (IPTG) in E. coli. After sonication, recombinant OAS-TL B proteins were purified from crude extract protein by HiTrap Chelating column (Cytiva) containing 50 mM $NiCl_2$ and eluted with buffer E (50 mM Tris-HCl pH 8.0, 250 mM NaCl, 400 mM Imidazol). Plant total soluble proteins were extracted as described above. OAS-TL enzyme activity was determined at 25 °C in a total volume of 100 μl containing 50 mM HEPES/KOH (pH 7.5), 5 mM $Na_2S$, 10 mM OAS (Bachem), 5 mM DTT and 1-2 μl of crude extract proteins or 1:100 diluted recombinant proteins.

## In vitro CSC dissociation capability assay

To investigate the OAS dissociation capability of the pCSC formed by SERAT2;1 and different alleles of OAS-TL B protein, a revised in vitro pull-down assay was performed as described previously[17]. Recombinant His-SERAT2;1, untagged OAS-TL B (M167A) and untagged OAS-TL B (S172N) were expressed in E. coli, respectively, as described above. Crude protein extract of His-SERAT2;1 expressing bacteria was added to a HiTrap Chelating column (Cytiva) containing 50 mM $NiCl_2$ and the column was then washed with Buffer W (50 mM Tris-HCl, pH 8.0, 250 mM NaCl, 80 mM Imidazol) and 10 mM OAS to remove the bacterial OAS-TL. Subsequently, the crude extract containing untagged OAS-TL B proteins was added to the column containing the His-SERAT2;1 protein. After specific binding of OAS-TL B to SERAT2;1, the OAS-TL B protein was specifically eluted with 10 mM OAS in Buffer W. After washing with Buffer W, the His-SERAT2;1 was final eluted with Buffer E (50 mM Tris-HCl, pH 8.0, 250 mM NaCl, 400 mM Imidazol). The fractions obtained from different elution steps were analyzed by SDS-PAGE followed by immunological detection of OAS-TLs using an anti-OAS-TL C antibody (against full-length Arabidopsis OAS-TL C, 1:2000 dilution) and a mouse anti-His-tag antibody (MA1-21315, Invitrogen, 1:2000 dilution).

## In vivo analyses of ABA response in guard cells with the pRAB18::GFP reporter

The pRAB18::GFP transgenic plants[28] were crossed with different genotypes and $F_3$ homozygous plants were used for analysis. Peels of pRAB18::GFP transgenic plants were floated on distilled water (pH 5.5) for 2 h under constant light and transferred to distilled water (pH 5.5) supplemented without (control) or with different effectors for different times as described above. Subsequently, guard cells were visualized with a confocal laser scanning microscope (Leica Stellaris 8) under excitation with 2% laser intensity. Green fluorescence (488 nm excitation/500 nm to 550 nm emission) for GFP reporter visualization, red fluorescence (488 nm excitation/600 nm to 750 nm emission) for chlorophyll visualization, and bright field images were obtained. All the

experiments were performed with identical settings. Stomata with both guard cells in focus were manually selected and image analysis was performed with the Fiji software (version 1.54 f)[69]. The mean gray values of GFP and chlorophyll were determined and GFP signals were normalized with chlorophyll signals.

### In vivo analyses of ABA in guard cells using the ABAleon2.1 sensor

Detection of the relative ABA level in the cytosol of guard cells after application of OPDA was performed by ABAleon2.1 as described previously[6]. Guard cells embedded in epidermal peels were treated with effectors as described. The ABAleon2.1 signal was visualized with a confocal laser scanning microscope (LSM510, Zeiss) using the following settings: Cyan fluorescence was measured in C1 channel (458 nm excitation, 475 nm to 500 nm window emission) for normalization. Yellow fluorescence upon energy transfer was measured in C2 channel (458 nm excitation and 525 nm to 550 nm window emission). Yellow fluorescence was also measured directly in C3 channel (514 nm excitation and 525 nm to 550 nm emission), and C3 was only used to assess overall image quality. The ratio of fluorescence resonance energy transfer (FRET) is calculated as the C2/C1 ratio, which was calculated via Fiji software (version 1.54 f)[69]. All the microscopy imaging was performed with identical settings on the same day.

### Water loss

To measure the loss of water from detached leaves, plants were grown on soil for 8 or 9 weeks under short-day conditions. Water loss experiments were performed four hours after the dark-light transition to ensure that the stomata were fully opened. Whole rosette leaves were detached from hypocotyl and placed on plates in a growth chamber (100 $\mu$mol m$^{-2}$ s$^{-1}$, 22 °C, 50% relative humidity). Leaves were weighed at the times indicated. The amount of water loss of plants was calculated by changes in fresh weight and presented as percentages of initial fresh weight.

### Stomatal conductance and chlorophyll fluorescence

Plants were grown on soil under short-day conditions (100 $\mu$mol m$^{-2}$ s$^{-1}$, 22 °C, 50% relative humidity). The stomatal conductance was measured with an SC-1 leaf porometer 4 h after the dark-light transition. The chlorophyll fluorescence ($F_V/F_M$) was quantified with an OS-30p+ Chlorophyll Fluorometer (Opti-Sciences) according to the manufacturer's instructions ($N = 9$, 3 leaves from 3 individuals).

### Drought stress survival rate

12-day-old seedlings of plants were transferred to pots containing the same amount of soil ($60 \pm 1$ g) and grown for 6–7 days with sufficient watering under short-day conditions (8.5 h light, 100 $\mu$mol m$^{-2}$ s$^{-1}$ at 22 °C and 15.5 h dark at 18 °C, 50% relative humidity). In each experiment, 96 biological replicates were analyzed, which grew in six individual pots (16 seedlings per pot). The 18 or 19 day-old plants were then subjected to drought by withholding water for 14–18 days followed by rewatering for 3 days. The survival rates of the seedlings were recorded for each genotype.

### Reporting summary

Further information on research design is available in the Nature Portfolio Reporting Summary linked to this article.

## Data availability

All data supporting the findings of this study are available in this paper. The plant materials used in this study are available from the corresponding author upon request. Source data are provided with this paper.

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

## Acknowledgements

We thank Prof. Rainer Hederich (University of Würzburg, Germany), Dr. E. Hornung (Georg-August-Universität Göttingen, Germany) and Prof. Roberto Solano (National Center for Biotechnolgy, Spain) for kindly providing mutants and Metabolomics Core Technology Platform of the Excellence Cluster "CellNetworks" (University of Heidelberg, [grant no. ZUK 40/2010–3009262]) for support with UPLC-based metabolite quantification. This work was supported by the Deutsche Forschungsgemeinschaft (DFG, German Research Foundation) with grants to Prof. Rüdiger Hell and Dr. Markus Wirtz (project IDs: 235736350, 452933265 and 544882710) and a postdoctoral fellowship from the Alexander von Humboldt Stiftung to S.K.S.

## Author contributions

R.H. and M.W. supervised the project. M.W. and S.K.S. designed experiments. S.K.S. conducted most of the experiments. S.K.S., N.A., H.C., H.R. and J.R.C.C. performed measurements of stomatal aperture. S.K.S. and H.C. performed ABA measurements in guard cells with the pRAB18::GFP reporter and stomatal conductance. V.V.U. performed ABA measurements in guard cells with the ABAleon2.1 sensor. R.H., M.W. and F.J.Z. provided funds and resources. S.K.S. prepared all figures and drafted the manuscript with input from M.W., M.W. and R.H. wrote and revised the manuscript with input from F.J.Z.

## Funding

## Competing interests

The authors declare no competing interests.
