## [Transparent Peer Review file · Nature Communications]

The plastid cysteine synthase complex regulates ABA biosynthesis and stomatal closure in Arabidopsis

Corresponding Author: Dr Markus Wirtz

Version 0:

Reviewer comments:

Reviewer #1

(Remarks to the Author)

Building on their previous paper published in *The Plant Cell* (Dec 2018) showing that the assimilation of sulfate into cysteine is essential for sulfate-induced abscisic acid (ABA) biosynthesis and stomatal closure in Arabidopsis, the authors further investigated the role of cysteine-synthase-complex in chloroplasts (pCSC) in response to soil-drying signals and high light stress conditions. This study integrates three drought-responsive axes (axis1: sulfate, axis2: CLE25 and axis3: oxylipin OPDA), components of which were partly reported before, and the hub of the axes were identified as pCSC. This study revealed the importance of pCSC in ABA production and stomatal closure for response to drought and high light stresses.

There are 3 major concerns about the study:

Firstly, the authors demonstrated the roles of drought-responsive axis1 (sulfate) and axis2 (CLE25) separately. However, the downstream signaling events connecting the CLE25 receptor BAM1/3 to pCSC remain unclear. Given that BAM1/3 likely phosphorylates a downstream target to transmit the signal, further clarification is needed. While the authors discussed beautifully on SERAT1 and SERAT2s, the potential role of SERAT3s, which localized in cytosol, should also be considered. The possible overlap between axis 1 and axis 2 remains unexplored. Could sulfate induce CLE25 expression or vice versa?

Secondly, the authors introduced high light stress into the "sensor-hub" model as a concurrent event with drought stress, which is reasonable. However, the definition of the high light stress in this study is vague. A dose-response analysis is required to conduct whether the varying light intensities effect the OPDA-pCSC-ABA signaling axis.

Thirdly, the authors mentioned that there's a novel way to genialize drought-resistant plants without growth penalty. Additional phenotypic evidence is needed, such as biomass accumulation, flowering time, and seed yield. Another aspect is the conservation and the activity of AtOAS-TL BS173N in other species, especially crops.

I think the authors need to fill the three major concerns listed above of the independency and potential crosstalk between the axes, the clearer definition of the high light stress and additional validation of the "no growth penalty" drought-resistant phenotype and conservation of the key point mutant. And the second section of the introduction need to be revised to clarify the key concepts.

Minor revision :

1. In result "pCSC is essential for drought stress-induced stomatal closure", it would be better to include the Cys content of WT and pCSC mutants(serat2.1 and oastlb).
2. The drought phenotypes of WT shown in Figure 1e and Supplementary Figure 1b exhibit notable discrepancies. Please verify and revise it.
3. The drought phenotypes of oastlb in Supplementary Figure 1b and Supplementary Figure 2c show relatively significant discrepancies. It is recommended to replace them with more consistent phenotypes to ensure data coherence and reliability.
4. In Supplementary Figure 3, the stomatal aperture of the opr3 under control treatment shows no significant difference compared to WT, which conflicts with the statement in the text: "The OPDA conversion mutant, opr3, accumulates OPDA and suffers from closed stomata". Please verify the data.
5. The "Materials and Methods section" contains several unit notation errors. Please verify and revise.
6. Reference formatting needs to be standardized, please check and revise.

Reviewer #2

(Remarks to the Author)

The authors investigated functions of plastid cysteine synthase complexes (pCSC) in ABA synthesis and stomatal closure and claimed that pCSC plays as a central hub for stress-induced regulation of ABA biosynthesis in guard cells and stomatal closure response, which is crucial for soil water limitation and high light stress. This topic is interesting. However, there are several concerns in this study.

Plastid include not only chloroplasts but also chromoplasts, leucoplasts, and so on. Is pCSC localized in only chloroplasts or in all plastids?

The authors incubated peels in 15 mM MgSO₄ for 3 h, 50 μM ABA for 1 h, 10 μM OPDA for 2 h, and 1 μM CLE25 for 3h. It is unclear that the obtained results are comparable. Why were incubation times different? The author should show time-course data and explain the reason. Furthermore, the authors tested 50 μM ABA for stomatal closure, which is too high. The authors should test 1 μM and 10 μM ABA. The authors should also show dose-dependency for all chemicals. The authors tested MgSO₄. Isn't there an effect of Mg²⁺ on ABA biosynthesis and stomatal closure?

Figure 1: Stomatal apertures are larger than other many reports. The authors should make it clear where they measured.

The authors measure stomatal apertures (n=50) in Figure 1b and (n=187-209) in Figure 1c. Why are the repeat counts different? It seems intentional. It should be same in all figures.

Figure 3d: the authors incubated the samples for 10 min. Why is it different? The authors should show time-course data?

Reviewer #3

(Remarks to the Author)

This study by Sun et al. uncovers a novel role of the plastid-localized cysteine synthase complex (pCSC) in regulating ABA biosynthesis and consequently controlling stomatal closure in *Arabidopsis thaliana*. The authors identify that the dynamic assembly of the pCSC, rather than its cytosolic or mitochondrial counterparts, is critical for initiating ABA biosynthesis in response to drought and high light stress. They discovered that three distinct stress signals, sulfate, CLE25, and OPDA, converge on the pCSC to regulate this process. Interestingly, OPDA also triggers pCSC formation through CYP20-3. In OPDA-overaccumulating mutants (*opr3*), stomata remain closed due to elevated ABA levels, but crossing these mutants with pCSC-deficient or CYP20-3-deficient lines reopens stomata, confirming the OPDA-pCSC-ABA signaling pathway. These signals trigger the assembly of the pCSC, enhancing cysteine synthesis within chloroplasts, which in turn supports ABA production specifically in guard cells.

The authors' findings suggesting that pCSC serves as a central hub for ABA biosynthesis and stomatal regulation are highly intriguing, and the proposed OPDA-pCSC-ABA signaling mechanism is compelling. However, the current data are insufficient to fully support this conclusion, and substantial revisions to the manuscript are required.

1. The authors should discuss reference 33 in more depth in the Introduction:

“Chang, Y. et al. Light-induced stomatal opening in *Arabidopsis* is negatively regulated by chloroplast-originated OPDA signaling. *Curr. Biol.* 33, 1071–1081.e5 (2023).”

This study was the first to demonstrate that OPDA contributes to stomatal closure under high light conditions. The novelty of the current manuscript lies in the proposed link between OPDA and ABA biosynthesis, and this context should be clearly articulated.

2. The authors should also discuss the reported role of OPDA-bound CYP20-3 in CSC formation (<https://doi.org/10.1093/jxb/erae421>) in the Introduction or main text.

3. Given the emphasis on OPDA and ABA as critical chemical signals, quantitative analyses of these molecules should be added at key points. Specifically:

•Lines 121–133: Include ABA accumulation data in Fig. 1e.

•Lines 171–175: Include data on OPDA accumulation.

•Lines 253–269: Provide ABA accumulation data for OAS-TL B(M167A) and OAS-TL B(S172N) overexpression lines.

4. The proposed relationship between OPDA, CYP20-3, and pCSC relies heavily on speculation and lacks sufficient experimental evidence. The authors should consider using the *jassy* mutant (www.pnas.org/cgi/doi/10.1073/pnas.1900482116), which is defective in OPDA export across the chloroplast membrane, to further support their claims.

5. The physical interaction between CYP20-3 and pCSC should be experimentally verified. Furthermore, the authors should test whether the presence or absence of OPDA affects this interaction.

6. The *opr3* mutant alone is insufficient to evaluate OPDA-specific effects (DOI: 10.1038/NCHEMBIO.2540). If possible, the *opr2-1 opr3-3* double mutant should be used.

7. To convincingly demonstrate that OPDA acts independently of JA-Ile, experiments using *coi1-1* or *jar1-1* mutants are necessary.

Version 1:

Reviewer comments:

Reviewer #1

(Remarks to the Author)

The authors have addressed my comments by performing additional experiments. I am satisfied with this revisions and have no further questions.

Reviewer #3

(Remarks to the Author)

The authors have addressed all of the concerns raised in the previous round of review with great care. They have also performed a substantial number of additional experiments, which strengthen and clarify the central claims of manuscript. The proposed OPDA–CYP20-3–pCSC–ABA signaling cascade constitutes a novel mechanism underlying stomatal closure and offers important new insights into plant stress response pathways. Overall, I believe this paper represents a valuable contribution to the field. I recommend the manuscript for publication.

Small correction:

PXA1 is also referred to as CTS or PED3 in the literature; therefore, appropriate references should be included.

Additional comments regarding the response to Referee #2:

Response 1

Comments: The authors have adequately responded to the reviewer's comments. In addition, regarding cysteine biosynthesis in plastids other than chloroplasts, I consider it unnecessary to include this topic in the present manuscript.

Response 2

Comments: Stomatal closure assays often differ in experimental conditions across studies and research groups. Therefore, the authors' results, obtained under conditions consistent with previous reports, are appropriate. Moreover, as stomatal closure occurs in a compound concentration-dependent manner, and comparative experiments with the oastlB mutant have also been conducted, the authors' conclusions are valid.

Response 3

Comments: The authors have adequately responded to the reviewer's comments.

Response 4

Comments: In stomatal closure experiments, the aperture of stomata may depend on measurement site and thus differ among studies and researchers. Then, the reviewer's point is reasonable. At the same time, the authors have addressed this issue by clearly indicating the measurement site. I would further request the following revisions:

1. Add to the figure legend that the typical aperture value for open stomata using this method (Batool et al., 2018; Savchenko et al., 2014; Chang et al., 2023) is 3–5 μm .
2. In Supplementary Fig. 1a, the measurement site is difficult to discern. Please replace it with an enlarged image of a single stoma in which the measurement site is clearly visible.

Response 5

Comments: In the Methods section, please clarify that multiple stomata can be obtained from the same sample, and provide a more detailed description of the procedure.

Response 6

Comments: The authors have adequately addressed the reviewer's comments. Stomatal closure induced by high-light treatment and that induced by compound treatment show completely different time courses. Since compound permeability also influences the response, it is experimentally appropriate to carefully consider treatment duration when administering compounds externally.

Responses to reviewers' comments

We would like to thank the reviewers for their positive and constructive comments to improve our manuscript. According to the reviewers' comments, we have performed several new experiments and revised the relevant text in the improved manuscript (highlighted in yellow color). Please find the point-by-point response to the reviewers concerns and suggestions below in blue color.

Reviewer #1 (Remarks to the Author):

Building on their previous paper published in *The Plant Cell* (Dec 2018) showing that the assimilation of sulfate into cysteine is essential for sulfate-induced abscisic acid (ABA) biosynthesis and stomatal closure in *Arabidopsis*, the authors further investigated the role of cysteine-synthase-complex in chloroplasts (pCSC) in response to soil-drying signals and high light stress conditions. This study integrates three drought-responsive axes (axis1: sulfate, axis2: CLE25 and axis3: oxylipin OPDA), components of which were partly reported before, and the hub of the axes were identified as pCSC. This study revealed the importance of pCSC in ABA production and stomatal closure for response to drought and high light stresses.

Response: The authors want to thank Reviewer 1 for the critical but fair evaluation of our manuscript.

There are 3 major concerns about the study:

Firstly, the authors demonstrated the roles of drought-responsive axis1 (sulfate) and axis2 (CLE25) separately. However, the downstream signaling events connecting the CLE25 receptor BAM1/3 to pCSC remain unclear. Given that BAM1/3 likely phosphorylates a downstream target to transmit the signal, further clarification is needed. **While the authors discussed beautifully on SERAT1 and SERAT2s, the potential role of SERAT3s, which localized in cytosol, should also be considered. The possible overlap between axis 1 and axis 2 remains unexplored. Could sulfate induce CLE25 expression or vice versa?**

Response: We thank reviewer 1 for this suggestion and tested the relevance of SERAT3;1 and SERAT3;2 for stress signals-induced stomata closure. Analysis of loss of SERAT3-function mutant (*serat3;1serat3;2* double mutant) revealed closure of the stomata like the wild type upon application of all mentioned stress signals. These results are in agreement with our central claim that only cysteine production by the chloroplast-localized CSC is critical for triggering ABA

production in guard cells. These results have been added as new **Supplementary Fig. 1** and **8** and in **Lines 107 and 173**.

With respect to a potential role crosstalk between CLE25 and sulfur metabolism we tested if sulfate administration to leaves induces transcription of CLE25 or if CLE25 application triggers transcriptional induction of the sulfate transporter (*Sultr*) genes. We found no indication of sulfate-induced CLE25 expression in roots and leaves, as well as CLE25-induced *Sultr* genes in leaves (**Supplementary Fig. 3a, b, d**). However, in roots, several *Sultr* genes slightly became up- or down-regulated at the transcript level upon CLE25 treatment (**Supplementary Fig. 3c**), indicating a potential regulation of sulfur metabolism by CLE25, which is discussed in the revised manuscript (**Lines 115-120**). Further experimentation might address this crosstalk at the protein level, but this potential crosstalk is out of the focus of this initial report on the importance of the pCSC for sensing long-distance drought-stress-related signals.

Secondly, the authors introduced high light stress into the “sensor-hub” model as a concurrent event with drought stress, which is reasonable. **However, the definition of the high light stress in this study is vague. A dose-response analysis is required to conduct whether the varying light intensities effect the OPDA-pCSC-ABA signaling axis.**

Response: We followed the suggestion of reviewer 1 and included a dose-response analysis to varying light conditions. Since 100 μ E was a suitable light intensity to which Arabidopsis plants were acclimated in our growth chamber system, we applied 300, 500, 1000, and 2000 μ E as mild and intense high-light stress conditions and tested the stomatal response in wild-type plants and the *oast1b* mutant. We found that mild high-light stress conditions also trigger stomata closure in a pCSC-dependent manner. Data on stomatal closure as a function of light intensity have been provided in **Supplementary Fig. 13c** and **Lines 226-227**.

Thirdly, the authors mentioned that there's a novel way to genialize drought-resistant plants without growth penalty. **Additional phenotypic evidence is needed, such as biomass accumulation, flowering time, and seed yield. Another aspect is the conservation and the activity of AtOAS-TL BS173N in other species, especially crops.**

Response: We followed the reviewer's suggestion and conducted an independent experiment to determine that biomass accumulation, flowering time, and seed yield in plants expressing AtOAS-

TL BM167A are unaffected (**Supplementary Fig. 19c-f, Line 289**). The importance of the chloroplast-localized pCSC for regulating sulfur metabolism has been previously evidenced in the crop rice (Sun *et al.*, 2021). However, the relevance of the chloroplast-localized pCSC for stress responses and regulation of sulfur assimilation in other plant species remains elusive.

References:

Sun, S.K., *et al.* (2021). A molecular switch in sulfur metabolism to reduce arsenic and enrich selenium in rice grain. *Nature Communications*, 12(1), 1392.

I think the authors need to fill the three major concerns listed above of the independency and potential crosstalk between the axes, the clearer definition of the high light stress and additional validation of the “no growth penalty” drought-resistant phenotype and conservation of the key point mutant. And the second section of the Introduction need to be revised to clarify the key concepts.

Response: We believe that the additional experiments we conducted have adequately addressed all of Reviewer 1’s concerns and further support our central claim (see detailed responses). Moreover, we have revised Section 2 of the Introduction to include additional background on ABA and cysteine biosynthesis, making the manuscript more accessible to non-expert readers.

Minor revision :

1. In result "pCSC is essential for drought stress-induced stomatal closure", it would be better to include the Cys content of WT and pCSC mutants (*serat2.1* and *oast1b*).

Response: The cysteine contents of the WT and the pCSC mutants (*serat2.1* and *oast1b*) are shown in the improved version of **Supplementary Fig. 4a** and **Lines 138-140**. Compared to WT, *serat2.1* and *oast1b* mutants showed either no difference or only a minimal increase in cysteine content in total leaves, which is consistent with the cysteine data for both mutants shown in Fig. 5a-b. The lack of pCSC subunits obviously caused compensatory production of cysteine in other subcellular compartments, resulting in a slightly enhanced foliar cysteine steady-state level.

2. The drought phenotypes of WT shown in Figure 1e and Supplementary Figure 1b exhibit notable discrepancies. Please verify and revise it.

Response: The reviewer is correct that the drought stress recovery experiments displayed substantial variability, depending on the time of re-watering the plants. We have repeated the drought recovery experiment and found consistently that mutants impaired in the pCSC were more sensitive to drought stress than the WT and mutants impaired in the cytosolic or mitochondrial CSC. We have replaced **Fig. 1e** with a more coherent set of data.

3. The drought phenotypes of *oast1b* in **Supplementary Figure 1b** (Supplementary Fig. 4b in the improved manuscript) and **Supplementary Figure 2c** (Supplementary Fig. 6d in the improved manuscript) show relatively significant discrepancies. It is recommended to replace them with more consistent phenotypes to ensure data coherence and reliability.

Response: As explained in the response to the previous questions, the recovery of the drought-stressed plants is strongly dependent on the time of re-watering after the onset of drought. This time point was slightly different between the independent repetitions of the experiments. We repeated the experiments and consistently found that the complementation of the *oast1b* mutant with OAS-TLB reverted the drought-sensitive phenotype of *oast1b*. We have replaced **Supplementary Fig. 6d** with a more coherent set of data.

4. In **Supplementary Figure 3** (Supplementary Fig. 7 in the improved manuscript), the stomatal aperture of the *opr3* under control treatment shows no significant difference compared to WT, which conflicts with the statement in the text: "The OPDA conversion mutant, *opr3*, accumulates OPDA and suffers from closed stomata". Please verify the data.

Response: We thank the reviewer for carefully reading our manuscript and pointing out this apparent contradiction. The reason for the open stomata phenotype of *opr3-1* in the epidermal peeling experiment is the pre-treatment with an opening buffer (50 mM KCl, 10 mM MES, pH 5.5, as indicated in the M&M section). This pre-treatment was applied to ensure that all effector treatments on stomata begin under the same condition (full opening of the stomata) allowing for comparison of the effector's impact on stomata from different genotypes. Even after pre-treatment with the opening buffer, the stomatal aperture of *opr3-1* is still significantly lower than that of WT (**Supplementary Fig. 7b**). We will explain this apparent contradiction by highlighting this treatment not only in the materials & methods section but also in the legend of the improved version of new **Supplementary Fig. 7b**. As shown in **Supplementary Fig. 14b**, the imprinting of

stomata embedded in leaves of the *opr3-1* mutant (without any pre-treatment) verified the published phenotype of constitutively more closed stomata in *opr3-1* (Savchenko *et al.*, 2014).

References:

Savchenko, T., *et al.* (2014). Functional convergence of oxylipin and abscisic acid pathways controls stomatal closure in response to drought. *Plant Physiology*, 164(3), 1151-1160.

5. The “Materials and Methods section” contains several unit notation errors. Please verify and revise.

Response: Done

6. Reference formatting needs to be standardized, please check and revise.

Response: Done

Reviewer #2 (Remarks to the Author):

The authors investigated functions of plastid cysteine synthase complexes (pCSC) in ABA synthesis and stomatal closure and claimed that pCSC plays as a central hub for stress-induced regulation of ABA biosynthesis in guard cells and stomatal closure response, which is crucial for soil water limitation and high light stress. This topic is interesting. However, there are several concerns in this study.

Response: We thank Reviewer 2 for acknowledging the importance of our study.

Plastid include not only chloroplasts but also chromoplasts, leucoplasts, and so on. Is pCSC localized in only chloroplasts or in all plastids?

Response: OAS-TLB and SERAT2;1 possess canonical targeting sequences for plastid localization. Consequently, these proteins should also be imported into chromoplasts and leucoplasts, which are predominantly present in generative tissues (petals, fruit pericarp cells) and the roots. Since these tissues lack or possess only very few stomata, cysteine production for ABA-triggered stomata closure in those plastid types is of minor importance for this study.

However, we agree with the reviewer that cysteine production in these plastid types may be relevant to the development and growth of roots and vegetative organs. We previously showed that the plastid-localized OAS-TL B alone is sufficient for normal root and flower development (Birke *et al.*, 2013). However, it is currently unclear if this is caused by the transport of reduced sulfur species from the leaves to these organs or by de novo cysteine synthesis in roots or generative organs. Due to the technical difficulties in dissecting the importance of the pCSC in those plastid types for de novo cysteine synthesis in tissues or the cysteine transport from other tissues, this intriguing research question must be addressed in a separate study.

References:

Birke, H., Heeg, C., Wirtz, M., & Hell, R. (2013). Successful fertilization requires the presence of at least one major O-acetylserine (thiol) lyase for cysteine synthesis in pollen of *Arabidopsis*. *Plant Physiology*, 163(2), 959-972.

The authors incubated peels in 15 mM MgSO₄ for 3 h, 50 μM ABA for 1 h, 10 μM OPDA for 2 h, and 1 μM CLE25 for 3h. It is unclear that the obtained results are comparable. Why were incubation times different? The author should show time-course data and explain the reason. Furthermore, the authors tested 50 μM ABA for stomatal closure, which is too high. The authors should test 1 μM and 10 μM ABA. The authors should also show dose-dependency for all chemicals.

Response: We thank the reviewer for carefully reading our manuscript. The selection of incubation times and concentrations of different effectors was based on previously published studies analyzing these effectors (Batool *et al.*, 2018; Savchenko *et al.*, 2014; Takahashi *et al.*, 2018). However, we fully understand the reviewer's concern and rigorously analyzed the impact of the effectors in time-course experiments and at different concentrations (**Supplementary Fig. 2a-f and Fig. 9a-b**). We found that, in WT plants, the degree of stomata closure depended on the treatment time, and all effectors could trigger significant but mild stomata closure within one hour (**Supplementary Fig. 2a-c and Fig. 9a**). Full stomata closure occurred after treatment with ABA (50 μM) for 1 hour, after treatment with sulfate (15 mM) for 2 hours and after treatment with CLE25 (1 μM) for 3 hours. Next, we tested the impact of different effector concentrations on stomata closure in the wild type and a pCSC-defective mutant (*oast1b*). As suggested by the reviewer, ABA also triggered stomata closure when 10 μM or 1 μM ABA was applied to the wild type or *oast1B* for one hour, which is not in conflict with any of our conclusions (**Supplementary Fig. 2d**).

Next, we tested a lowered concentration of sulfate (from 15 to 1 mM) and CLE25 (from 1 μ M to 0.01 μ M). We found that lower concentrations of both effectors also triggered stomata closure in the wild type after application for two and three hours, respectively. Most importantly, the *oast1b* mutant did not close the stomata after application of high sulfate or OPDA concentration (used in the study) or low sulfate and OPDA concentrations (**Supplementary Fig. 2e-f**). The same type of response was observed in time-course experiments and by application of varying concentrations of the effector OPDA (**Supplementary Fig. 9a-b**). Thus, we conclude that the effector concentrations and incubation times applied in this study are suitable for analyzing the physiological responses to these effectors, which is in agreement with published manuscripts (Batool *et al.*, 2018; Savchenko *et al.*, 2014; Takahashi *et al.*, 2018).

To further investigate the role of ABCG40 in the ABA import into guard cells, we conducted a time-course experiment using 50 μ M ABA on stomata in epidermal peels of the wild type and loss-of-function mutants (*abcg40-1*, *abcg40-2*, Kang *et al.*, 2010). In all experiments, we aimed to use a very high ABA concentration to ensure that stomata in the wild type are fully closed. While the wild type fully closed the stomata after one hour of ABA treatment, stomata of both *abcg40* mutants remained open during the entire experiment (last time point 3 hours, **Supplementary Fig. 16a**).

References:

Batool, S., *et al.* (2018). Sulfate is incorporated into cysteine to trigger ABA production and stomatal closure. *The Plant Cell*, 30(12), 2973-2987.

Kang, J., *et al.* (2010). PDR-type ABC transporter mediates cellular uptake of the phytohormone abscisic acid. *Proceedings of the National Academy of sciences*, 107(5), 2355-2360.

Savchenko, T., *et al.* (2014). Functional convergence of oxylipin and abscisic acid pathways controls stomatal closure in response to drought. *Plant Physiology*, 164(3), 1151-1160.

Takahashi, F., *et al.* (2018). A small peptide modulates stomatal control via abscisic acid in long-distance signalling. *Nature*, 556(7700), 235-238.

The authors tested MgSO₄. Isn't there an effect of Mg²⁺ on ABA biosynthesis and stomatal closure?

Response: A substantial impact of the here applied Mg^{2+} concentration on stomata closure has already been excluded by treating stomata with $MgCl_2$ in our previous study on sulfate-induced stomata closure (Figure 1 of Batool *et al.*, 2018). However, we have repeated this experiment with another Mg^{2+} source to exclude any impact of Mg^{2+} on stomatal movement. These data (**please see the Figure below**) reinforce our initial findings and demonstrate that, in short-term treatment (3 h), only sulfate (Na_2SO_4 or $MgSO_4$) triggers stomatal closure, while $Mg(NO_3)_2$ has no impact. Since this dataset is rather confirmatory, we prefer not to include it as the supplemental figure.

References:

Batool, S., *et al.* (2018). Sulfate is incorporated into cysteine to trigger ABA production and stomatal closure. *The Plant Cell*, 30(12), 2973-2987.

Fig. Stomatal apertures of epidermal peels treated with different ions (15 mM).

Isolated stomata in epidermal peels of wild type were treated with water (control), 15 mM $Mg(NO_3)_2$, 15 mM $MgSO_4$, 15 mM Na_3PO_4 , or 15 mM Na_2SO_4 for 3 h, respectively. Data are shown as means \pm SD (n = 50). Statistical differences were analyzed by one-way ANOVA followed by Tukey's test. **** $P < 0.0001$. ns, no significant difference.

Figure 1: Stomatal apertures are larger than other many reports. The authors should make it clear where they measured.

Response: The reviewer is correct that reported stomatal apertures in the literature vary substantially. This discrepancy is predominantly caused by two factors: firstly, analyzing stomata from leaves of different growth stages, which vary in the development stage of stomata, and secondly, by the different methodologies used to determine and display stomatal apertures. In method 1, the width of the stomata aperture is determined and reported. Typical values for the opened stomata aperture, as determined by this method, range between 3-5 μm (Batool *et al.*, 2018; Savchenko *et al.*, 2014; Chang *et al.*, 2023), which is precisely what we have found by applying this method on fully developed leaves. In the improved version of **Supplementary Figure 1a**, we have defined the position for measuring the stomata aperture in a representative example. The second widely applied method determines the width and length of the stomata and reports the ratio of width to length (Takahashi *et al.*, 2018; Förster *et al.*, 2019). Typical values for stomata apertures reported by this method (width/length ratio) are <1 .

References:

Batool, S., *et al.* (2018). Sulfate is incorporated into cysteine to trigger ABA production and stomatal closure. *The Plant Cell*, 30(12), 2973-2987.

Chang, Y., *et al.* (2023). Light-induced stomatal opening in Arabidopsis is negatively regulated by chloroplast-originated OPDA signaling. *Current Biology*, 33(6), 1071-1081.

Förster, S., *et al.* (2019). Wounding-induced stomatal closure requires jasmonate-mediated activation of GORK K⁺ channels by a Ca²⁺ sensor-kinase CBL1-CIPK5 complex. *Developmental Cell*, 48(1), 87-99.

Savchenko, T., *et al.* (2014). Functional convergence of oxylipin and abscisic acid pathways controls stomatal closure in response to drought. *Plant Physiology*, 164(3), 1151-1160.

Takahashi, F., *et al.* (2018). A small peptide modulates stomatal control via abscisic acid in long-distance signalling. *Nature*, 556(7700), 235-238.

The authors measure stomatal apertures (n=50) in Figure 1b and (n=187-209) in Figure 1c. Why are the repeat counts different? It seems intentional. It should be same in all figures.

Response: We thank the reviewer for pointing out the potential threat to the credibility of our statements due to the use of different numbers of replicates in those experiments. We will explain in the figure caption of improved Figure 1 that the dynamic impact of the effectors on stomata in epidermal peels was determined after a defined time, which restricted the time for data acquisition

in this experimental setup. Thus, we decided to determine the apertures of 50 stomata in this experimental setup (Figure 1b). When we imprinted leaves (Figure 1c), we were able to take more pictures from the entire imprint, including many imprinted and, therefore, fixed stomata, allowing us to increase the number of quantified stomata up to around 200.

Figure 3d: the authors incubated the samples for 10 min. Why is it different? The authors should show time-course data?

Response: The reviewer is correct in that the duration of the high light exposure triggers rapid stress-induced stomata closure in intact leaves, which differs from the time of effector-triggered closure of stomata embedded in the epidermal peels. This difference is caused by the nature of the highly diverse signals and the different experimental setup. In both cases, we could confirm the time scales previously reported for these signals (high light → range of minutes (Devireddy *et al.*, 2018, 2020) and chemical effectors like ABA → range of hours in the epidermal peel system (Malcheska *et al.*, 2017, Kang *et al.*, 2010)). In addition, we followed the suggestion of Reviewer 2 and rigorously assessed the time of the effector treatment by performing detailed time-course data. These novel data are presented in the new **Supplementary Figs. 2a-c (Lines 111 + 115), 9a (Lines 174-176), and 13b (Lines 226-228)**, confirming that the experiments were conducted at a suitable time for all tested treatments.

References:

Devireddy, A. R., *et al.* (2018). Coordinating the overall stomatal response of plants: rapid leaf-to-leaf communication during light stress. *Science Signaling*, 11(518), eaam9514.

Devireddy, A. R., *et al.* (2020). Coordinated and rapid whole-plant systemic stomatal responses. *New Phytologist*, 225(1), 21-25.

Kang, J., *et al.* (2010). PDR-type ABC transporter mediates cellular uptake of the phytohormone abscisic acid. *Proceedings of the National Academy of sciences*, 107(5), 2355-2360.

Malcheska, F., *et al.* (2017). Drought-enhanced xylem sap sulfate closes stomata by affecting ALMT12 and guard cell ABA synthesis. *Plant Physiology*, 174(2), 798-814.

Reviewer #3 (Remarks to the Author):

This study by Sun *et al.* uncovers a novel role of the plastid-localized cysteine synthase complex (pCSC) in regulating ABA biosynthesis and consequently controlling stomatal closure

in *Arabidopsis thaliana*.

The authors identify that the dynamic assembly of the pCSC, rather than its cytosolic or mitochondrial counterparts, is critical for initiating ABA biosynthesis in response to drought and high light stress. They discovered that three distinct stress signals, sulfate, CLE25, and OPDA, converge on the pCSC to regulate this process. Interestingly, OPDA also triggers pCSC formation through CYP20-3. In OPDA-overaccumulating mutants (*opr3*), stomata remain closed due to elevated ABA levels, but crossing these mutants with pCSC-deficient or CYP20-3-deficient lines reopens stomata, confirming the OPDA-pCSC-ABA signaling pathway. These signals trigger the assembly of the pCSC, enhancing cysteine synthesis within chloroplasts, which in turn supports ABA production specifically in guard cells.

The authors' findings suggesting that pCSC serves as a central hub for ABA biosynthesis and stomatal regulation are highly intriguing, and the proposed OPDA-pCSC-ABA signaling mechanism is compelling.

Response: We thank the reviewer for the positive evaluation of our data and research strategy.

However, the current data are insufficient to fully support this conclusion, and substantial revisions to the manuscript are required.

1. The authors should discuss reference 33 in more depth in the Introduction:

“Chang, Y. et al. Light-induced stomatal opening in *Arabidopsis* is negatively regulated by chloroplast-originated OPDA signaling. *Curr. Biol.* 33, 1071–1081.e5 (2023).”

This study was the first to demonstrate that OPDA contributes to stomatal closure under high light conditions. The novelty of the current manuscript lies in the proposed link between OPDA and ABA biosynthesis, and this context should be clearly articulated.

Response: We thank Reviewer 3 for his/her fair evaluation of our study and his/her critical comment on the novelty of the OPDA-induced stomata closure upon high-light stress. The authors believe that the latter comment is not supported by the cited reference 33, which we mentioned in the discussion of the old manuscript version. Chang and coworkers demonstrate that OPDA is a negative regulator of the light-induced stomata opening during the dark-to-light transition. Light-induced stomata opening is a fundamentally different process from the high-light stress-induced stomata closure that occurs during the day in response to light fluctuations. We carefully re-analysed whether Chang and coworkers performed any high-light stress experiments and found no high-light stress-related data in the main body text or the supplemental information.

However, we followed the suggestion of Reviewer 3 and provided a more detailed explanation of the findings of Chang and coworkers in the Introduction (**Lines 77-78**). As **Chang *et al.* found a negative role of OPDA in the dark-to-light transition-induced stomatal opening**, the data support the **role of OPDA as a positive regulator of stomatal closure under high light stress**, as revealed in our manuscript. We also agree with reviewer 3 that the proposed mechanistic link between OPDA and ABA biosynthesis adds to the novelty of our study. However, the central claim of our study is that the pCSC integrates water limitation signals transmitted from roots to shoots (sulfate and CLE25) and local stress signals, such as OPDA (triggered by high-light stress), to facilitate stomatal closure in an ABA-dependent manner.

2. The authors should also discuss the reported role of OPDA-bound CYP20-3 in CSC formation (<https://doi.org/10.1093/jxb/erae421>) in the Introduction or main text.

Response: We followed the suggestion of reviewer 3 and discussed the established role of CYP20-3 as an OPDA sensor that triggers pCSC formation, as reported by Adhikari *et al.* (2025), in the improved version of the Discussion section (**Lines 346-354**).

References:

Adhikari, A., *et al.* (2025). OPDA signaling channels resource (e^-) allocation from the photosynthetic electron transfer chain to plastid cysteine biosynthesis in defense activation. *Journal of Experimental Botany*, 76(2), 594-606.

3. Given the emphasis on OPDA and ABA as critical chemical signals, quantitative analyses of these molecules should be added at key points. Specifically:

- Lines 121–133: Include ABA accumulation data in Fig. 1e.
- Lines 171–175: Include data on OPDA accumulation.
- Lines 253–269: Provide ABA accumulation data for OAS-TL B(M167A) and OAS-TL B(S172N) overexpression lines.

Response: We followed the suggestion of Reviewer 3 (comment for lines 253-269) and included foliar ABA levels on top of the measurements of cytosolic ABA levels in guard cells by the well-established ABA reporter pRAB18-GFP (**Fig. 4d and 4g**). In the case of ectopic expression of OAS-TL B(M167A) and OAS-TL B(S172N) by the 35S-promoter in all leaf-cells, including the stomata, we found a specific accumulation of ABA after the expression of OAS-TL B(M167A),

which causes stable formation of the plastid CSC (**Fig. 4d**). Expression OAS-TL B(S172N), failing to form a stable pCSC complex, resulted in wild-type ABA levels (**Fig. 4d**). These data support the central claim that pCSC formation is critical for ABA biosynthesis in guard cells.

As suggested by reviewer 3 (comment for lines 121-133), we also included foliar ABA levels after water limitation. These novel data are included in **Supplementary Figs. 4c-e and 5a-c** of the revised manuscript version. Soil drying led to a massive accumulation of foliar ABA, likely due to the induction of vascular ABA biosynthesis (Endo *et al.*, 2008; Kuromori *et al.*, 2014). However, this massive ABA accumulation in leaves occurred after soil drying-induced stomatal closure and was unaffected by the absence of pCSC (**Supplementary Fig. 4c-e and 5a-c**), strongly suggesting that the massive accumulation of ABA in leaves is rather a later and secondary signal to maintain stomatal closure. This later signal is essential for a successful drought response because the initial closure of stomata by upregulating ABA biosynthesis in guard cells creates CO₂ deficiency in the mesophyll, resulting in the formation of an ABA-independent stomatal opening signal that must be counterbalanced (Engineer *et al.*, 2016). These novel data are included and explained in the results section (**Lines 141-146**).

We followed the suggestion of reviewer 3 (comment for lines 171-175) and included OPDA measurements from total leaves of *opr3-1* and *opr3-3* in **Supplementary Fig. 11d-e**. However, consistent with previous reports (Stintzi *et al.*, 2001; Chehab *et al.*, 2011; Savchenko *et al.*, 2014), we did not detect a massive accumulation of OPDA in total leaf extracts of *opr3-1* mutants under non-stressed conditions. Although Chang *et al.* reported that *opr3-3* significantly accumulates OPDA (1.5 ng/mg in WT and 3 ng/mg in *opr3-3*) (Chang *et al.*, 2023), we did not observe a statistically significant accumulation of OPDA in total leaf extracts of *opr3-3* when compared to wild type (**Supplementary Fig. 11e**). A possible explanation for this difference could be different growth conditions, different metabolite extraction protocols, and the unusual harvesting time (after dark-to-light transition) applied in Chang *et al.*, 2023. However, the fact that we do not observe a significant accumulation of OPDA in total leaf extract does not mean that OPDA cannot accumulate in the chloroplasts of guard cells, which contribute only 0.25% to the volume of leaf 6 (first adult leaf, Tolleter *et al.*, 2024).

References:

Chang, Y., *et al.* (2023). Light-induced stomatal opening in Arabidopsis is negatively regulated by chloroplast-originated OPDA signaling. *Current Biology*, 33(6), 1071-1081.

Chehab, E. W., *et al.* (2011). Intronic T-DNA insertion renders *Arabidopsis opr3* a conditional jasmonic acid-producing mutant. *Plant Physiology*, 156(2), 770-778.

Endo, A., *et al.* (2008). Drought induction of *Arabidopsis* 9-cis-epoxycarotenoid dioxygenase occurs in vascular parenchyma cells. *Plant Physiology*, 147(4), 1984-1993.

Engineer, C. B., *et al.* (2016). CO₂ sensing and CO₂ regulation of stomatal conductance: advances and open questions. *Trends in Plant Science*, 21(1), 16-30.

Kuromori, T., *et al.* (2014). Intertissue signal transfer of abscisic acid from vascular cells to guard cells. *Plant Physiology*, 164(4), 1587-1592.

Savchenko, T., *et al.* (2014). Functional convergence of oxylipin and abscisic acid pathways controls stomatal closure in response to drought. *Plant Physiology*, 164(3), 1151-1160.

Stintzi, A., *et al.* (2001). Plant defense in the absence of jasmonic acid: the role of cyclopentenones. *Proceedings of the National Academy of Sciences*, 98(22), 12837-12842.

Tolleter, D., *et al.* (2024). The *Arabidopsis* leaf quantitative atlas: a cellular and subcellular mapping through unified data integration. *Quantitative Plant Biology*, 5, e2.

4. The proposed relationship between OPDA, CYP20-3, and pCSC relies heavily on speculation and lacks sufficient experimental evidence. The authors should consider using the *jassy* mutant (www.pnas.org/cgi/doi/10.1073/pnas.1900482116), which is defective in OPDA export across the chloroplast membrane, to further support their claims.

Response: We appreciate the reviewer's suggestion. To provide even more evidence for our claim, we have applied the *pxa1-1* mutant, which is deficient in the OPDA import into peroxisomes and substantially accumulates OPDA (Park *et al.*, 2013, Dave *et al.*, 2011). Like the *opr3* mutants, the *pxa1-1* mutant showed decreased stomatal aperture (**Supplementary Fig. 11c**). Please note that the *pxa1-1* mutant is defective in seed germination, which has never been reported for the *jassy* mutant, and that the seed germination inhibition phenotype of *ped3-3* (allele of *pxa1-1* in *Ler* background) can be rescued by the absence of ABI5, an ABA responsive transcription factor (Kanai *et al.*, 2010).

The JASSY protein is localized at the outer membrane of the plastid envelope (Guan *et al.*, 2019). Consequently, OPDA would accumulate predominantly in the intermembrane space of plastids in the *jassy* mutants, but OPDA accumulation has so far not been reported for the *jassy* mutant. In

contrast to *opr3* mutants, the *jassy* mutant is not impaired in self-fertilization (Guan *et al.*, 2019), a condition that can be rescued in *opr3* mutants by JA application (Sanders *et al.*, 2000; Stintzi & Browse, 2000). This means that the *jassy* mutant is less impaired in JA biosynthesis than *opr3-1* and *pxa1-1* and has no problem producing sufficient amounts of JA from OPDA under non-stressed conditions. The latter fact demonstrates that a backup OPDA transporting system exists at the outer membrane of plastids, enabling efficient OPDA export from the intermembrane space of plastids into the cytosol under non-stressed conditions in the *jassy* mutant. This backup system also explains the nearly unaffected steady-state level of JA in leaves of the *jassy* mutant under non-stressed conditions (Guan *et al.*, 2019). In contrast, *opr3* and *pxa1-1* mutants all display significantly lower levels of JA under non-stressed conditions (Chehab *et al.*, 2011, Chini *et al.*, 2018, Savchenko *et al.*, 2014, Theodoulou *et al.*, 2005). However, this backup system can obviously not efficiently transport the massive amounts of OPDA produced in plastids in response to diverse stresses, explaining the stress-related phenotypes of *jassy* (Guan *et al.*, 2019).

Based on these facts and the almost wild type-like phenotype of the *jassy* mutant, we did not expect a substantial trapping and accumulation of OPDA in plastids under non-stressed conditions. In agreement, we did not observe stomata closure in the *jassy* mutant. These data are integrated into the improved version of **Supplementary Fig. 11b** and interpreted in the result section of the main body text (**Lines 199-207**).

References:

Chehab, E. W., *et al.* (2011). Intronic T-DNA insertion renders *Arabidopsis opr3* a conditional jasmonic acid-producing mutant. *Plant Physiology*, 156(2), 770-778.

Chini, A., *et al.* (2018). An OPR3-independent pathway uses 4, 5-didehydrojasmonate for jasmonate synthesis. *Nature Chemical Biology*, 14(2), 171-178.

Dave, A., *et al.* (2011). 12-Oxo-phytodienoic acid accumulation during seed development represses seed germination in *Arabidopsis*. *The Plant Cell*, 23(2), 583-599.

Guan, L., *et al.* (2019). JASSY, a chloroplast outer membrane protein required for jasmonate biosynthesis. *Proceedings of the National Academy of Sciences*, 116(21), 10568-10575.

Kanai, M., Nishimura, M., & Hayashi, M. (2010). A peroxisomal ABC transporter promotes seed germination by inducing pectin degradation under the control of ABI5. *The Plant Journal*, 62(6), 936-947.

Park, S., *et al.* (2013). The α/β hydrolase CGI-58 and peroxisomal transport protein PXA1 coregulate lipid homeostasis and signaling in *Arabidopsis*. *The Plant Cell*, 25(5), 1726-1739.

Sanders, P. M., *et al.* (2000). The *Arabidopsis* DELAYED DEHISCENCE1 gene encodes an enzyme in the jasmonic acid synthesis pathway. *The Plant Cell*, 12(7), 1041-1061.

Savchenko, T., *et al.* (2014). Functional convergence of oxylipin and abscisic acid pathways controls stomatal closure in response to drought. *Plant Physiology*, 164(3), 1151-1160.

Stintzi, A., & Browse, J. (2000). The *Arabidopsis* male-sterile mutant, opr3, lacks the 12-oxophytodienoic acid reductase required for jasmonate synthesis. *Proceedings of the National Academy of Sciences*, 97(19), 10625-10630.

Theodoulou, F. L., *et al.* (2005). Jasmonic acid levels are reduced in COMATOSE ATP-binding cassette transporter mutants. Implications for transport of jasmonate precursors into peroxisomes. *Plant Physiology*, 137(3), 835-840.

5. The physical interaction between CYP20-3 and pCSC should be experimentally verified. Furthermore, the authors should test whether the presence or absence of OPDA affects this interaction.

Response: The physical interaction between OPDA and CYP20-3 as well as CYP20-3 and the pCSC is well established by independent research groups (Dominguez-Solis *et al.*, 2008, Park *et al.*, 2013, Adhikari *et al.*, 2025), as indicated by Reviewer 3 in question 2. These reports unambiguously demonstrate the interaction between CYP20-3 and SERAT2;1, which triggers pCSC formation (see also response to question 2). The authors believe that the duplication of independently confirmed facts will dilute the central message of this manuscript, which already suffers from a high data load (20 supplementary figures).

References:

Dominguez-Solis, J. R., *et al.* (2008). A cyclophilin links redox and light signals to cysteine biosynthesis and stress responses in chloroplasts. *Proceedings of the National Academy of Sciences*, 105(42), 16386-16391.

Park, S. W., *et al.* (2013). Cyclophilin 20-3 relays a 12-oxo-phytyldienoic acid signal during stress responsive regulation of cellular redox homeostasis. *Proceedings of the National Academy of Sciences*, 110(23), 9559-9564.

Adhikari, A., *et al.* (2025). OPDA signaling channels resource (e⁻) allocation from the photosynthetic electron transfer chain to plastid cysteine biosynthesis in defense activation. *Journal of Experimental Botany*, 76(2), 594-606.

6. The *opr3* mutant alone is insufficient to evaluate OPDA-specific effects (DOI: 10.1038/NCHEMBIO.2540). If possible, the *opr2-1 opr3-3* double mutant should be used.

Response: We followed the suggestion of reviewer 3 and ordered the *opr2-1* and *opr3-3* single mutants from the European Arabidopsis Stock Centre. We tested stomata closure in the single *opr2-1* and *opr3-3* mutants. Like *opr3-1* (used in this study), *opr3-3* displayed constitutively closed stomata under non-stressed conditions (**Supplementary Fig. 11b**). Remarkably, the stomata of *opr2-1* mutants were open like in the wild type (**Supplementary Fig. 11b**). These data reveal a consistent correlation between stomata closure and blockage of the main OPDA to JA conversion pathway in Arabidopsis, strongly suggesting that OPDA accumulation triggers stomata closure. In this context, we want to emphasize that we have crossed *opr3-1* with mutants defective in the OPDA receptor CYP20-3, the pCSC (*serat2;1*, and *oast1b*), or ABA biosynthesis (*aba3-1*). Inhibition of the OPDA-pCSC-ABA signaling axis in these *opr3,1* containing double mutants reopened the stomata, providing direct genetic evidence that all components of this axis are essential for stomata closure in the *opr3-1* single mutant (**Fig. 2e, Supplementary Fig. 11g**). On top of that we have crossed the ABA-reporter *pRAB18::GFP* into *opr3-1* and showed that ABA sensing is activated in guard cells of *opr3;1* (**Fig. 2f**). The ABA sensing vanished in *opr3-1 cyp20-3* and *opr3-1 serat2;1* double mutants, demonstrating that ABA production in *opr3* was dependent on CYP20-3 and the pCSC (**Fig. 2f**). We also crossed *opr3-1* with *coi1-t* (lacking the ability to sense JA-Ile, Mosblech *et al.*, 2011), the *opr3-1 coi1-t* double mutant did not reopen the stomata, demonstrating that JA-Ile sensing is not contributing to stomata closure in *opr3-1* (**Supplementary Fig. 11f**).

References:

Mosblech, A., Thurow, C., Gatz, C., Feussner, I., & Heilmann, I. (2011). Jasmonic acid perception by COI1 involves inositol polyphosphates in Arabidopsis thaliana. *The Plant Journal*, 65(6), 949-957.

7. To convincingly demonstrate that OPDA acts independently of JA-Ile, experiments using *coi1-1* or *jar1-1* mutants are necessary.

Response: We agree with reviewer 3 that it is critical to test if OPDA will close stomata in the absence of JA-Ile formation (*jar1*) or JA-Ile sensing (*coi1*). We don't have *jar1-1* or *coi1-1* mutant lines in our hands, but we have applied other full-loss-of-function alleles of JAR1 and COI1 to test the importance of JA-Ile production (*jar1-11*) and JA-Ile sensing (*coi1-t* and *coi1-16*, which is a weak *coi1* allele).

The *coi1-t* allele is a full knock-out mutant of COI1, resulting in a sterile phenotype that cannot be rescued by spraying with methyl-jasmonate, as reported for the *coi1-1* mutant (Mosblech *et al.*, 2011, p. 951). The *jar1-11* is a total loss of JAR1, because of a T-DNA insertion in exon 3 of the gene, and shows the same phenotype as *jar1-1* (Suza and Staswick, 2008).

We thus hope that the application of these mutants sufficiently addresses Reviewer 3's concern. The impact of OPDA on the epidermal peels of both mutants and *coi1-16* is shown in **Supplementary Fig. 7c** of the improved manuscript. In agreement with a JA-Ile independent function of OPDA in stomata closure, OPDA triggered stomata closure in mutants lacking the formation or sensing of JA-Ile.

References:

Mosblech, A., *et al.* (2011). Jasmonic acid perception by COI1 involves inositol polyphosphates in *Arabidopsis thaliana*. *The Plant Journal*, 65(6), 949-957.

Suza, W. P., & Staswick, P. E. (2008). The role of JAR1 in jasmonoyl-L-isoleucine production during *Arabidopsis* wound response. *Planta*, 227(6), 1221-1232.

Responses to reviewers' comments

Reviewer #1 (Remarks to the Author):

The authors have addressed my comments by performing additional experiments. I am satisfied with this revisions and have no further questions.

Response: We thank reviewer 1 for his/her efforts to improve our study.

Reviewer #3 (Remarks to the Author):

The authors have addressed all of the concerns raised in the previous round of review with great care. They have also performed a substantial number of additional experiments, which strengthen and clarify the central claims of manuscript. The proposed OPDA–CYP20-3–pCSC–ABA signaling cascade constitutes a novel mechanism underlying stomatal closure and offers important new insights into plant stress response pathways. Overall, I believe this paper represents a valuable contribution to the field. I recommend the manuscript for publication.

Response: We appreciate the reviewer's positive comments regarding the revised manuscript and thank the reviewer for carefully reading and improving our manuscript with his/her valuable suggestions.

Small correction:

PXA1 is also referred to as CTS or PED3 in the literature; therefore, appropriate references should be included.

Response: We followed the suggestion of reviewer 3 and included references for PXA1/CTS/PED3 in line 207 (as ref. 37-39).

Additional comments regarding the response to Referee #2:

Response 1

Comments: The authors have adequately responded to the reviewer's comments. In addition, regarding cysteine biosynthesis in plastids other than chloroplasts, I consider it unnecessary to include this topic in the present manuscript.

Response: We thank reviewer 2 for his/her evaluation of the presented data.

Response 2

Comments: Stomatal closure assays often differ in experimental conditions across studies and research groups. Therefore, the authors' results, obtained under conditions consistent

with previous reports, are appropriate. Moreover, as stomatal closure occurs in a compound concentration-dependent manner, and comparative experiments with the oast1B mutant have also been conducted, the authors' conclusions are valid.

Response: We appreciate the reviewer's positive comments regarding the revised manuscript.

Response 3

Comments: The authors have adequately responded to the reviewer's comments.

Response: We appreciate the reviewer's positive comments regarding the revised manuscript.

Response 4

Comments: In stomatal closure experiments, the aperture of stomata may depend on measurement site and thus differ among studies and researchers. Then, the reviewer's point is reasonable. At the same time, the authors have addressed this issue by clearly indicating the measurement site. I would further request the following revisions:

1. Add to the figure legend that the typical aperture value for open stomata using this method (Batool et al., 2018; Savchenko et al., 2014; Chang et al., 2023) is 3–5 μm .
2. In Supplementary Fig. 1a, the measurement site is difficult to discern. Please replace it with an enlarged image of a single stoma in which the measurement site is clearly visible.

Response: We thank reviewer 2 for these helpful suggestions. We added the typical aperture value in figure legend of Supplementary Fig. 1a and replaced the old image of a single stomata with an enlarged version in "panel a" of Supplementary Fig. 1.

Response 5

Comments: In the Methods section, please clarify that multiple stomata can be obtained from the same sample, and provide a more detailed description of the procedure.

Response: We thank the reviewer for carefully reading our manuscript raise this suggestion. We defined in the improved Method section that we used peels from leaves of at least 4 individual plants for each treatment (Line 459).

Response 6

Comments: The authors have adequately addressed the reviewer's comments. Stomatal closure induced by high-light treatment and that induced by compound treatment show completely different time courses. Since compound permeability also influences the

response, it is experimentally appropriate to carefully consider treatment duration when administering compounds externally.

Response: We appreciate the reviewer's evaluation of our revision and want to thank the reviewer for the fair reviewing process.